# LM²otifs: An Explainable Framework for Machine-Generated Texts Detection

## Abstract

The impressive ability of large language models to generate natural text across various tasks has led to critical challenges in authorship authentication. Although numerous detection methods have been developed to differentiate between machine-generated texts (MGT) and human-generated texts (HGT), the explainability of these methods remains a significant gap. Traditional explainability techniques often fall short in capturing the complex word relationships that distinguish HGT from MGT. To address this limitation, we present LM²otifs, a novel explainable framework for MGT detection. Inspired by probabilistic graphical models, we provide a theoretical rationale for the effectiveness. LM²otifs utilizes eXplainable Graph Neural Networks to achieve both accurate detection and interpretability. The LM²otifs pipeline operates in three key stages: first, it transforms text into graphs based on word co-occurrence to represent lexical dependencies; second, graph neural networks are used for prediction; and third, a post-hoc explainability method extracts interpretable motifs, offering multi-level explanations from individual words to sentence structures. Extensive experiments demonstrate the comparable performance of LM²otifs. The empirical evaluation of the extracted explainable motifs confirms their effectiveness in differentiating HGT and MGT. Furthermore, qualitative analysis reveals *distinct and visible linguistic fingerprints* characteristic of MGT.

## 1 Introduction

Large Language Models (LLMs) have made remarkable progress in recent years, demonstrating the ability to generate text based on prompt instructions. Models like ChatGPT (OpenAI, 2022), Llama (Touvron et al., 2023), and Claude-3 (Anthropic, 2024) have shown impressive capabilities in writing (Yuan et al., 2022a), coding (Zhang et al., 2024c), and question answering (Zhuang et al., 2023). However, these advances raise serious concerns about content authenticity, including fake news (Ahmed et al., 2021), plagiarism (Lee et al., 2023), and misinformation (Chen & Shu, 2024). Given that humans struggle to identify machine-generated texts (MGT) (Gehrmann et al., 2019b), developing reliable detectors to distinguish between MGT and human-generated texts (HGT) has become essential.

Existing LLM detectors (Yang et al., 2024; Nguyen-Son et al., 2024; Guo et al., 2024b; Chang et al., 2024) are broadly categorized as white-box and black-box approaches. White-box approaches, exemplified by DetectLLM (Su et al., 2023), analyze the probabilities of the output token to identify distinguishing characteristics (Yu et al., 2024). In contrast, black-box methods (Guo et al., 2024b; Soto et al., 2024; Zhang et al., 2024b; Nguyen-Son et al., 2024) achieve detection without access to the LLM's internal workings. Despite their effectiveness, significant challenges persist in creating detectors that are both robust and explainable (Wu et al., 2025). Furthermore, these methods typically only output a binary classification. However, practical applications demand supporting evidence, such as the need for the determination of originality. However, existing explainability techniques for these detectors are inadequate. Traditional methods like Integrated Gradients (Sundararajan et al., 2017) are computationally prohibitive for LLM-based detectors, and while attention mechanisms (Jain & Wallace, 2019; Wiegreffe & Pinter, 2019) excel at capturing local dependencies, they may face challenges in identifying global patterns crucial for an LLM. Consequently, developing an explainable detector solution is critical and timely.

The fundamental architecture of modern LLM builds upon the principle of autoregressive next-token prediction, which models the joint probability distribution of a sequence as $P(s_1, s_2, \cdots, s_T) \approx \prod_{t=1}^{T} P_\theta(s_t|s_{1:t-1})$, where $\theta$ is the (trainable) model parameter, $s_i$ is the word/token at the $i$th position, and $T$ is bounded by the context length (Radford et al., 2019; Bengio et al., 2000). Following this notion, in MGT detection, current methods typically treat the input as sequential data, and measure the distance between its posterior distribution and reference distributions for MGT and HGT samples — for instance by estimating the Kullback-Leibler (KL) divergence. This often requires substantial computational resources and large sample sizes. However, an intuitive and efficient alternative, probabilistic graphical models (PGM) (Bishop & Nasrabadi, 2006; Koller, 2009), to model conditional probabilities, has been largely overlooked. From the perspective of PGM, while generation tasks require that LLM operate based on probability graphs which accurately approximate the ground-truth posterior distribution, detection tasks only require constructing and analyzing probability graphs that are sufficiently discriminative for the underlying detection task. With sufficient sample data, building such graphs is straightforward. Furthermore, by analyzing the mechanism between sequence-based detectors and graph-based detectors, we provide the advantage of graph-based detectors in theory. In practice, PGM has advantages in terms of explainability, inference speed, and detection accuracy.

Drawing inspiration from PGM, we introduce a novel explainable framework, LM$^2$OTIFS. Beyond classifying input text as either MGT or HGT, LM$^2$OTIFS generates explanatory motifs that justify its detection outcome. LM$^2$OTIFS consists of three key parts: i) Graph Construction, ii) MGT Detection, and iii) Explainable Motifs Extraction. In the first stage, we leverage the word co-occurrence techniques to capture the lexical dependencies. To extract meaningful patterns at multiple levels (e.g., words and phrases), we integrate mainstream eXplainable Graph Neural Networks (XGNNs) to generate these motifs. To validate the effectiveness of our PGM-inspired approach, we empirically demonstrate that LM$^2$OTIFS achieves competitive performance with state-of-the-art MGT detection methods, including both supervised and zero-shot approaches. Following eXplainable AI (XAI) protocols, we verify the effectiveness of LM$^2$OTIFS. Our results indicate that the generated explainable motifs significantly outperform the baseline in terms of interpretability. The main contributions of this paper are summarized as follows:

★ We highlight the problem of missing evidence support in MGT detection. We introduce LM$^2$OTIFS, an explainable framework for MGT detection that integrates co-occurrence graphs with XGNN techniques for both accurate detection and explainable motifs extraction.

★ We provide a theoretical analysis of the rationale and advantages of employing Graph Neural Network(GNN) for this task, drawing insights from the perspective of PGM.

★ We conduct comprehensive experiments on diverse datasets, validating the effectiveness of LM$^2$OTIFS in MGT detection. Our analysis following XAI protocols supports the correctness of the extracted explainable motifs.

## 2 Preliminary

**MGT Detection.** The MGT detection problem can be formulated as a classification task. Take an example of a binary hypothesis testing task. Given a pair of training sets,

$$\mathcal{T}_h = \{\mathbf{S}_{h,i} = (S_{h,i,1}, S_{h,i,2}, \cdots, S_{h,i,L_i})\}_{i \in |\mathcal{T}_h|},$$
$$\mathcal{T}_m = \{\mathbf{S}_{m,i} = (S_{m,i,1}, S_{m,i,2}, \cdots, S_{m,i,L'_i})\}_{i \in |\mathcal{T}_m|},$$

consisting of human-generated and machine-generated text sequences, respectively, drawn from the distributions[1] $P_h$ and $P_m$, the objective is to classify a newly observed text sequence $\mathbf{S}_o$ as either human-generated or machine-generated. A detection mechanism is a function $f : (\mathcal{T}_h, \mathcal{T}_m, \mathbf{S}_o) \mapsto \widehat{Y}$, where $\widehat{Y} \in \{0, 1\}$, the index 0 represents the null hypothesis (human generated) and 1 represents the alternative hypothesis (machine generated). Notably, the function $f$ takes input of $\mathbf{S}_o$ and $\mathcal{T}_h, \mathcal{T}_m$ are the support samples. In training-based methods, $\mathcal{T}_h, \mathcal{T}_m$ are used to train models, while in zero-shot methods, they are used to design the function, such as log rank information in DetectLLM (Su et al., 2023). The detection error is quantified by the risk function $P(f(\mathcal{T}_h, \mathcal{T}_m, \mathbf{S}_o) \neq Y)$, where $Y \in \{0, 1\}$ denotes the ground-truth hypothesis label.

---

[1]The length of the observed text sequences is not fixed and can be modeled as a random variable. This variability is implicitly captured in the distributions $P_h$ and $P_m$.

**Probabilistic Graphical Models**. PGM offers an efficient framework for representing probabilistic models, incorporating insightful properties such as conditional independence. Given a graph $G = \{\mathcal{V}, \mathcal{E}\}$, the nodes $\mathcal{V}$ correspond to random variables, and the links $\mathcal{E}$ capture probabilistic dependencies between these variables. For example, given a sequence of three tokens $\mathbf{S} = (s_1, s_2, s_3)$, the joint distribution is $P(s_1, s_2, s_3) = P(s_3|s_1, s_2)P(s_2|s_1)P(s_1)$. This can be represented using a graph with $\mathcal{V} = \{s_1, s_2, s_3\}$ and $\mathcal{E} = \{(s_1, s_2), (s_1, s_3), (s_2, s_3)\}$. More generally, for any sequence of tokens, a PGM can be constructed to represent the probabilistic dependencies among tokens.

**Node Classification**. A graph $G$ consists of a set of nodes $\mathcal{V} = \{v_1, v_2, \cdots, v_n\}$, where $n \in \mathbb{N}$, and a set of edges $\mathcal{E} \subseteq \mathcal{V} \times \mathcal{V}$. The adjacency matrix $\boldsymbol{A} \in \{0, 1\}^{n \times n}$ encodes the graph edges, where $A_{i,j} = \mathbb{1}((v_i, v_j) \in \mathcal{E})$. Each node may be associated with a feature vector, collectively represented by the matrix $\boldsymbol{X} \in \mathbb{R}^{n \times d}$, where the $i$-th row is the feature vector associated with the $i$-th node, and $d \in \mathbb{N}$ is the dimension. Each node $v$ is related to a label $Y_v \in \mathcal{Y}$, where $\mathcal{Y}$ is the collection of possible labels. In this work, we reformulate the author detection problem as a node classification task. This reformulation is elaborated on in the subsequent sections. The objective in node classification is to train a classifier $f : (\mathbf{G}, \boldsymbol{X}, v) \mapsto \widehat{Y}_v$, which, given a graph $\mathbf{G}$, node feature matrices $\boldsymbol{X}$, and a node index $v$, produces an estimate $\widehat{Y}_v$ of the node label $Y_v$. The accuracy of the classifier is defined as $P_{V, \mathbf{G}, \boldsymbol{X}, Y_V}(f(V, \mathbf{G}, \boldsymbol{X}) \neq Y_V)$, where $V$ is uniformly distributed over $\mathcal{V}$, and $\mathbf{G}, \boldsymbol{X}, Y_V$ follow a joint distribution $P_{\mathbf{G}, \boldsymbol{X}, Y_V}$.

**Post-hoc Explainable Graph Neural Networks**. Given a graph or node classification task, the goal of XGNN is to find an explanation function $\Psi(\cdot)$, which maps the input graph $G$ to a *minimal* and *sufficient* explanation subgraph $G_{exp}$. Minimality restricts the size of the explanatory subgraph and is enforced by the constraint $|G_{exp}| \leq s \cdot |G|$, where $|G|$ denotes the number of edges in $G$ and $s \in [0, 1]$ is the size parameter. Sufficiency is quantified by the KL divergence term $d_{KL}(P_{Y|G, \boldsymbol{X}, V} || P_{Y|G_{exp}, \boldsymbol{X}, V})$. The explainer is optimally sufficient if it minimizes the KL divergence subject to minimality constraints. That is, given $s \in [0, 1]$, an *optimal* explainer $\Psi^*$ is defined as:

$$\Psi^*(G) = \underset{\Psi : |G_{exp}| \leq s|G|}{\arg\min} \; d_{KL}(P_{Y|G, \boldsymbol{X}, V} || P_{Y|G_{exp}, \boldsymbol{X}, V}) \tag{1}$$

## 3 THEORETICAL ANALYSIS

As discussed in the prequel, prior works in MGT detection, such as Fast-DetectGPT (Bao et al., 2024), have employed sequential data models to design detection mechanisms. Drawing inspiration from TextGCN, we formulate the MGT detection problem using a graph-based approach where both tokens and documents are represented as nodes. Building upon this foundation, we demonstrate that GNN-based detectors achieve strictly improved detection accuracy compared to such approaches. This section provides theoretical justifications for this claim. The subsequent sections provide further verification through empirical analysis over several benchmark datasets.

We formally define a class of baseline *empirical sequential-based* (ESB) detectors that capture the essential characteristics of existing approaches. An ESB detector operates in two steps. First, it uses the human-generated training set $\mathcal{T}_h$ to construct the empirical conditional distribution estimates $\widehat{P}_h(s_t|s_{1:t-1})$ for human-generated text sequences, where $t \in [T]$, and $T$ is a hyperparameter capturing the maximum context length. Similarly, the empirical estimates $\widehat{P}_m(s_t|s_{1:t-1})$ are computed based on the machine generated training set $\mathcal{T}_m$. In the second step, the detector uses (a potentially trainable) mapping $g_s : ((\widehat{P}_h(s_t|s_{1:t-1}), \widehat{P}_m(s_t|s_{1:t-1}))_{t \in [T]}, \mathbf{S}_o) \mapsto \widehat{Y}$, where $\mathbf{S}_o$ is the to-be-classified sequence. An ESB detector is completely characterized by the mapping $g_s(\cdot)$. We denote the collection of ESB detectors by $\mathcal{F}_{\text{ESB}}$. We introduce the class of PGB MGT detectors. A PGB detector operates on a specially constructed graph with two types of nodes: token nodes and text sequence nodes (Yao et al., 2019). Formally, let $\mathcal{V} = \mathcal{S} \cup \mathcal{D}$ denote the complete node set, where

$$\mathcal{S} = \{s | \exists \mathbf{S} \in \mathcal{T}_h \cup \mathcal{T}_m, i \in [|\mathbf{S}|] : s_i = s\},$$
$$\mathcal{D} = \{\mathbf{S} | \mathbf{S} \in \mathcal{T}_h \cup \mathcal{T}_m \cup \{\mathbf{S}_o\}\}.$$

Here, $\mathcal{S}$ represents the set of all unique tokens in either human or machine-generated texts, and $\mathcal{D}$ comprises all text sequences from both sources and the to-be-classified text.

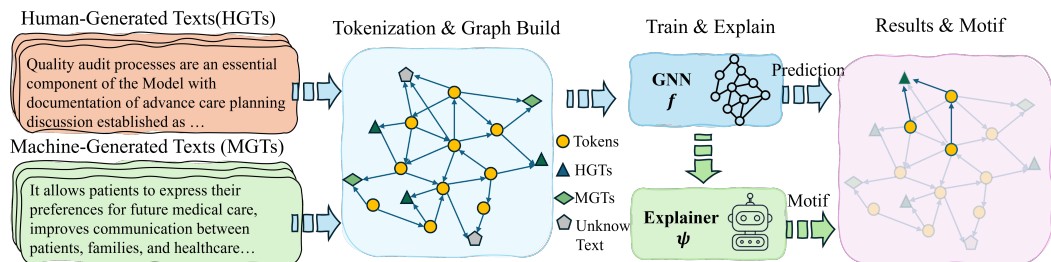

Figure 1: Overall pipeline of our framework, including tokenization, graph building, detector training, and motifs extraction.

The edge structure of the graph captures both token co-occurrences and token-sequence relationships. Two tokens $s_i, s_j \in \mathcal{S}$ are connected if they co-occur in at least $\lambda$ sequences within $\mathcal{T}_h \cup \mathcal{T}_m$, where $\lambda$ is a hyperparameter. Additionally, each token node is connected to sequence nodes containing that token. Edge weights are defined by two distinct functions. For token-token edges $(s_i, s_j)$, the PGB first computes embedding vectors for each token using

$$e_\ell : \mathcal{J}_\ell(s_i) \times (\mathcal{I}_{\ell,j}(s_i))_{j \in \mathcal{J}_\ell(s_i)} \mapsto \mathbf{e}_{\ell,i}, \quad \ell \in \{h, m\},$$

where for each token $s_i$, the set $\mathcal{J}_\ell(s_i) = \{j | s_i \in \mathbf{S}_{\ell,j}\}$ indexes the sequences containing $s_i$, while $\mathcal{I}_{\ell,j}(s_i) = \{k | S_{\ell,j,k} = s_i\}$ indexes the positions where $s_i$ appears in sequence $\mathbf{S}_{\ell,j}$. The token-token edge weights is then computed as $A_t(\mathbf{e}_{h,i}, \mathbf{e}_{m,i}, \mathbf{e}_{h,j}, \mathbf{e}_{m,j})$, where $\mathbf{e}_{h,i}$ and $\mathbf{e}_{m,i}$ are the embeddings from human and machine-generated texts, respectively. For token-sequence edges $(s, \mathbf{S})$, the weight is simply $A_s(N_{s|\mathbf{S}})$, where $N_{s|\mathbf{S}}$ counts occurrences of token $s$ in sequence $\mathbf{S}$. Examples of these edge weight functions $A_t(\cdot)$ and $A_s(\cdot)$ are provided in equation 2 and used in our empirical evaluations.

Token nodes are initialized with one-hot features and sequence nodes with all-zeros features. The GNN operates by several rounds of message passing among connected nodes. The PGB detector applies $K$ rounds of message passing over the constructed graph, where at each round, node embeddings are updated based on messages received from neighboring nodes. After $K$ iterations, the detector computes the final node embeddings, denoted by $\mathbf{h}^{(K)}$. The classification output is obtained via a function $g_p : (\mathbf{h}^{(K)}, \mathbf{S}_o) \mapsto \widehat{Y}$ that maps the collection of node embeddings to the binary decision $\widehat{Y}$. A PGB detector is completely characterized by the tuple $(K, \lambda, e_h, e_m, A_t, A_s, g_p)$. We denote the collection of PGB detectors by $\mathcal{F}_{\text{PGB}}$.

The following theorem shows that the PGB class of detectors strictly subsumes the ESB class in terms of achievable detection accuracy.

**Theorem 3.1.** *For every ESB detector $f_{ESB} \in \mathcal{F}_{ESB}$, there exists a PGB detector $f_{PGB} \in \mathcal{F}_{PGB}$ such that the detection accuracy of $f_{PGB}$ matches that of $f_{ESB}$, i.e.,*

$$P(f_{PGB}(\mathcal{T}_h, \mathcal{T}_m, \mathbf{S}_o) = Y) = P(f_{ESB}(\mathcal{T}_h, \mathcal{T}_m, \mathbf{S}_o) = Y),$$

*for all pairs of probability distributions $(P_h, P_m)$. Furthermore, the PGB class of detectors strictly improves upon the ESB class in terms of detection accuracy. That is, for any fixed set of hyperparameters $T, K, \lambda$, there exists $(P_h, P_m)$ and $f_{PGB} \in \mathcal{F}_{PGB}$ for which:*

$$P(f_{PGB}(\mathcal{T}_h, \mathcal{T}_m, \mathbf{S}_o) = Y) > \max_{f_{ESB} \in \mathcal{F}_{ESB}} P(f_{ESB}(\mathcal{T}_h, \mathcal{T}_m, \mathbf{S}_o) = Y),$$

The proof is provided in **Appendix A**.

## 4 METHODOLOGY

In this section, drawing upon the theoretical foundations of PGM in the prequel, we present the practical implementation of our probabilistic graph-based (PGB) detector framework, $\text{LM}^2\text{OTIFS}$. Our implementation encompasses three key components: graph construction based on token co-occurrences, GNN-based authorship detection, and explainable motif extraction. The complete pipeline of $\text{LM}^2\text{OTIFS}$ is illustrated in Figure 1.

## 4.1 GRAPH CONSTRUCTION

Following our PGB framework, we implement an efficient graph construction method based on co-occurrence principles from TextGCN (Yao et al., 2019). Our pipeline consists of two stages. In the first stage, we capture the dependencies among words/tokens. As shown in Figure 2, a word-dependency graph (solid lines) is constructed using a sliding window. In the second stage, we add document nodes and connect them to the corresponding words (dashed lines) that appear in the document. During testing, we similarly add test-document nodes to the existing word-dependency graph. Finally, our graph consists of two types of nodes representing tokens and documents, corresponding to the node sets $\mathcal{S}$ and $\mathcal{D}$. As specified in our framework, tokens are initialized with one-hot features and documents with zero vectors.

To construct edges that capture textual relationships, we consider both document-token connections and token co-occurrences. The adjacency matrix $A$ is defined as:

$$A_{ij} = \begin{cases} 1 & i,j \text{ are token}, \text{PMI}(i,j) > 0 \\ 1 & j \text{ is document}, i \text{ is token in } j \\ 1 & i = j \\ 0 & \text{otherwise} \end{cases}, \quad (2)$$

where $\text{PMI}(i,j) = \log\frac{p(i,j)}{p(i)p(j)}$, point-wise mutual information, is used to determine significant token co-occurrences. Here, $p(i)$ represents the frequency of the $i$-th token within a fixed-length sliding window, and $p(i,j)$ denotes the co-occurrence frequency of tokens $i$ and $j$. As discussed in Section 3, in the most general sense, the edge weights may be continuous-valued, and generated using a learnable function. However, our experimental evaluation shows that the above binary-valued edge weights are sufficient for reliable detection.

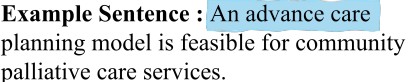

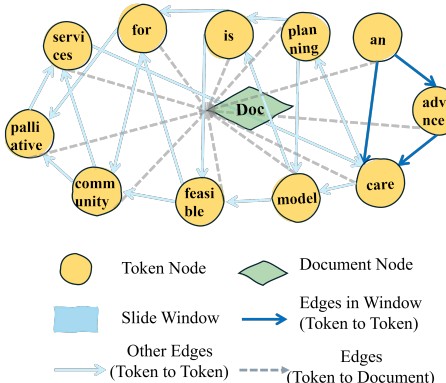

Figure 2: An example of graph construction with a fixed sliding window size 3.

## 4.2 GNN DETECTION

Having constructed the graph structure, we implement the detection mechanism outlined in our framework through a GNN architecture. For a given text sequence $\mathbf{S}_o$, our goal is to learn a function $f$ that determines whether the text is machine-generated or human-authored. This corresponds to the PGB detector operating over $K$ message passing rounds. Each GNN layer implements one round of message passing, with the update rule:

$$a_v^{(l)} = \text{AGG}^{(l)}\left(h_u^{(l-1)} : u \in \mathcal{N}(v)\right),$$

$$h_v^{(l)} = \text{COMBINE}^{(l)}\left(h_v^{(l-1)}, a_v^{(l)}\right),$$

where $a_v^{(l)}$ represents the aggregated message at $l$-th layer, $h_v^{(l)}$ is the node feature, $\mathcal{N}(v)$ denotes the neighbors of node $v$, and $\text{AGG}(\cdot)$ and $\text{COMBINE}(\cdot)$ are the regular aggregation and combination functions in GNNs, following the definition from previous work (Xu et al., 2019). After $K$ layers, we obtain the final node embeddings $\boldsymbol{H}$. For classification, we apply a softmax function to the final embeddings to obtain prediction probabilities $\boldsymbol{Z} = \text{softmax}(\boldsymbol{H})$. The model is trained by minimizing the cross-entropy loss over labeled document nodes:

$$\mathcal{L} = -\sum_{d \in \mathcal{Y}_D} \sum_{\ell \in \{h,m\}} Y_{d\ell} \ln Z_{d\ell}, \quad (3)$$

where $\mathcal{Y}_D$ represents the set of document nodes in the training set and $Y_{d\ell}$ is the ground-truth label,. While our goal focuses on binary classification (human-authored vs. machine-generated) in this paper, the framework naturally extends to scenarios with multiple classes, such as texts generated by different language models.

### 4.3 Explainable Motifs Extraction

Beyond detection accuracy, our framework provides interpretable insights through the extraction of distinguishing motifs between machine-generated and human-authored texts. While existing detection methods often operate as black boxes (Guo et al., 2024b), our graph-based approach naturally enables the identification of characteristic patterns through subgraph structures (Koller, 2009). Drawing inspiration from graph analysis techniques (Luo et al., 2020), we transform the interpretability challenge into a subgraph identification problem, where meaningful token dependencies in our constructed graph serve as distinguishing motifs. These motifs capture characteristic patterns of word usage and dependencies that differentiate between human and machine-generated content (Kim et al., 2024), providing insights beyond simple token-level statistics.

Specifically, we formulate a practical optimization objective using cross-entropy loss and explicit size constraints. The objective function balances the prediction accuracy of the explanation subgraph against its complexity:

$$\Psi^*(\cdot) = \operatorname*{arg\,min}_{\Psi: G \mapsto G_{exp}} \operatorname{CE}(Y; f(G_{exp})) + \lambda |G_{exp}| \qquad (4)$$

where $\operatorname{CE}(Y; f(G_{exp}))$ measures how well the explainer preserves the model's prediction capability, $|G_{exp}|$ denotes the size of the explanation subgraph, and $\lambda$ controls the trade-off between explanation fidelity and complexity. This formulation is an approximation of the theoretical requirements from Equation 1, where the cross-entropy term ensures sufficiency and the size penalty enforces minimality. The optimization is performed through gradient descent, with the edge weights of $G_{exp}$ being learned continuously and then discretized through thresholding.

## 5 Related Work

**AI-generated text Detection**. Detecting machine-generated texts approaches can be categorized into three main categories. The first category focuses on watermarking LLM-generated content (Chang et al., 2024; Ajith et al., 2024; Yang et al., 2023; Wu et al., 2024; Molenda et al., 2024). Most watermarking methods operate in a white-box setting, where researchers can modify the decoding process or token distribution directly (Ajith et al., 2024; Wu et al., 2024; Molenda et al., 2024). The black-box setting can be achieved by implementing post-processing modules to embed watermarks (Chang et al., 2024; Yang et al., 2023). The second category encompasses training-based detection methods that leverage trained neural networks (Guo et al., 2024b; Solaiman et al., 2019; Zhang et al., 2024b; Kim et al., 2024; Soto et al., 2024). OpenAI developed GPT-2 detectors using RoBERTa (Liu, 2019) as their foundation model (Solaiman et al., 2019). Additionally, researchers have explored fine-tuning language models specifically for detection purposes (Li et al., 2023; Koike et al., 2024; Guo et al., 2023; Zhang et al., 2024a). The third category consists of zero-shot detection methods (Nguyen-Son et al., 2024; Zeng et al., 2024; Yang et al., 2024; Tian et al., 2024; Ma & Wang, 2024), which utilize existing tools like LLMs without additional training. For example, SimLLM (Nguyen-Son et al., 2024) generates comparative text samples to identify machine-generated content through similarity analysis. R-Detect (Song et al., 2025) suggests a non-parametric kernel relative test to check if a text's distribution is closer to HGT than MGT.

**Explainable LLMs & GNNs**. Large language models often function as black-box systems, presenting inherent risks for downstream applications (Zhao et al., 2024). To address this limitation, researchers have developed various explanation methods (Wu et al., 2020; Li et al., 2016; Enguehard, 2023; Chen et al., 2023), which can be divided into local and global approaches. Local explanation methods aim to illuminate how an LLM arrives at predictions for specific inputs (Wu et al., 2020; Li et al., 2016; Chen et al., 2023). For example, the leave-one-out technique represents a fundamental approach to measuring input feature importance (Wu et al., 2020; Li et al., 2016). Global explanation methods focus on understanding how specific model components operate, including hidden layers and language model mechanisms. For instance, researchers have tracked attention layers to extract semantic information (Wu et al., 2020). SASC (Singh et al., 2023) employs pre-trained models to generate explanations for various LLM components.

Various approaches have emerged for extracting subgraph explanations using GNNs (Yuan et al., 2022b; Lin et al., 2021; Fang et al., 2023; Xie et al., 2022; Chen et al., 2024). These methods can be

Table 1: Detection performance comparisons on HGT and MGT based on ACC. The best and second-best results are shown in bold and underlined, respectively. YSC represents the combination of the Yelp, Essay, and Creative datasets.

| Method | M4 | | | | RAID | | | | YSC | | | |
|---|---|---|---|---|---|---|---|---|---|---|---|---|
| | DaV. | Coh. | Dol. | Blo. | Lla. | GT4 | MPT | Mis. | Son. | Opu. | Gem. | Avg. |
| Likelihood Solaiman et al. (2019) | 0.69 | 0.87 | 0.66 | 0.54 | 0.79 | 0.75 | 0.50 | 0.65 | 0.80 | 0.83 | 0.74 | 0.71 |
| Rank Gehrmann et al. (2019b) | 0.51 | 0.54 | 0.53 | 0.53 | 0.53 | 0.53 | 0.51 | 0.52 | 0.52 | 0.52 | 0.52 | 0.52 |
| LogRank Ippolito et al. (2019) | 0.67 | 0.88 | 0.72 | 0.62 | 0.80 | 0.74 | 0.46 | 0.66 | 0.76 | 0.79 | 0.72 | 0.71 |
| Entropy Gehrmann et al. (2019b) | 0.62 | 0.61 | 0.53 | 0.53 | 0.62 | 0.62 | 0.52 | 0.63 | 0.72 | 0.74 | 0.64 | 0.62 |
| NPR Su et al. (2023) | 0.63 | 0.67 | 0.55 | 0.59 | 0.79 | 0.66 | 0.54 | 0.65 | 0.70 | 0.63 | 0.54 | 0.63 |
| LRR Su et al. (2023) | 0.77 | 0.75 | 0.74 | 0.77 | 0.87 | 0.71 | 0.55 | 0.71 | 0.74 | 0.72 | 0.53 | 0.72 |
| DetectGPT Mitchell et al. (2023) | 0.48 | 0.57 | 0.48 | 0.59 | 0.67 | 0.59 | 0.46 | 0.53 | 0.62 | 0.58 | 0.57 | 0.56 |
| Fast-DetectGPT Bao et al. (2024) | 0.81 | **0.98** | 0.90 | 0.54 | 0.94 | 0.85 | 0.48 | 0.64 | 0.85 | 0.88 | 0.76 | 0.78 |
| DNAGPT Yang et al. (2024) | 0.53 | 0.74 | 0.53 | 0.50 | 0.68 | 0.66 | 0.39 | 0.54 | 0.62 | 0.64 | 0.65 | 0.59 |
| Binoculars Ma & Wang (2024) | 0.83 | 0.97 | 0.90 | 0.66 | **0.98** | 0.92 | 0.58 | 0.71 | 0.88 | 0.91 | 0.81 | 0.83 |
| Glimpse Bao et al. (2025) | 0.74 | 0.94 | 0.69 | 0.61 | 0.88 | 0.77 | 0.68 | 0.77 | 0.85 | 0.85 | 0.76 | 0.69 |
| GPTZero Tian, Edward (2023) | 0.74 | 0.80 | 0.61 | 0.53 | 0.65 | 0.60 | 0.54 | 0.55 | 0.69 | 0.71 | 0.54 | 0.63 |
| RoBERTa-QA Guo et al. (2023) | 0.83 | 0.94 | 0.74 | 0.51 | 0.77 | 0.70 | 0.56 | 0.56 | 0.79 | 0.87 | 0.80 | 0.73 |
| Radar Hu et al. (2023) | 0.76 | 0.77 | 0.65 | 0.63 | 0.68 | 0.69 | 0.64 | 0.72 | 0.80 | 0.83 | 0.78 | 0.72 |
| DeTeCtive Guo et al. (2024b) | 0.90 | 0.85 | 0.90 | 0.92 | 0.96 | 0.97 | **0.92** | 0.88 | 0.94 | 0.91 | 0.86 | 0.91 |
| LM$^2$OTIFS | **0.95** | 0.97 | **0.91** | **0.98** | **0.98** | **1.00** | 0.90 | **0.91** | **0.99** | **0.99** | **0.91** | **0.95** |

categorized into several groups. Gradient-based traditional approaches, including SA (Baldassarre & Azizpour, 2019) and Grad-CAM (Pope et al., 2019), leverage gradient information to derive explanations. Model-agnostic techniques encompass three main categories. First, perturbation-based methods such as GNNExplainer (Ying et al., 2019), PGExplainer (Luo et al., 2020), and ReFine (Wang et al., 2021b) identify important features and subgraph structures through systematic perturbations. Second, surrogate methods (Vu & Thai, 2020; Duval & Malliaros, 2021) approximate local predictions using surrogate models to generate explanations. Third, generation-based approaches (Yuan et al., 2020; Shan et al., 2021; Wang & Shen, 2023) employ generative models to produce both instance-level and global-level explanations.

# 6 EXPERIMENTS

We conduct extensive experiments to evaluate LM$^2$OTIFS across two aspects: MGT detection performance, and explainable motifs effectiveness. For MGT detection, we compare LM$^2$OTIFS against state-of-the-art supervised and zero-shot detectors on multiple benchmark datasets in both **in-domain** and **cross-domain** aspects. To validate our explainable motifs, we follow the (Hooker et al., 2019; Zheng et al., 2025) to use Most Relevant First (MoRF) and Least Relevant First (LeRF) to verify the effectiveness. Due to the limitation of space, we provide **ablation studies**, **time complexity**, implementation, and **motifs statistical analysis** in **Appendix C**.

## 6.1 SETUPS

**Datasets**. Following established benchmarks in MGT detection (Yang et al., 2024; Zeng et al., 2024), we evaluate LM$^2$OTIFS on six comprehensive datasets: **HC3** (Guo et al., 2023), **M4** (Wang et al., 2024), and **RAID** (Dugan et al., 2024), **Yelp** (Mao et al., 2024), **Creative**, **Essay** (Verma et al., 2023; Guo et al., 2024a). We select four domains in each dataset: open-qa, wiki-csai, medicine, and finance in HC3; wiki-how, reddit, peerread, and arxiv in M4; and recipes, book summaries, poetry, and IMDB reviews in RAID. The HC3 dataset only contains **ChatGPT**-generated text. While in M4 and RAID, there are several kinds of LLM-generated texts. In this paper, we also consider language models: **DaVinci**(DaV.), **Cohere**(Coh.), **Dolly**(Dol.), and **BloomZ**(Blo.) in M4, **Llama2**(Lla.), **GPT-4**(GT4), **MPT**, and **Mistral**(Mis.) in RAID. In Yelp, Creative, and Essay, we consider three LLMs, **Claude3-Sonnet**(Son.), **Claude-3-Opus**(Opu.), and **Gemini-1.0-Pro**(Gem.). The dataset details are available in **Appendix B.1**.

**Baselines**. We consider the training-based and zero-shot detection methods, such as DetectLLM (Su et al., 2023), DeTeCtive (Guo et al., 2024b), DNAGPT (Yang et al., 2024). To make a unified com-

parison protocol, both training-based and zero-shot methods use pre-trained models for comparison. For the explainable evaluation, we introduce a simple random motif for comparison due to the lack of existing methods. The detailed information is provided in Appendix B.2.

**Implementation**. Our experiments are based on a three-layer GCN architecture, and GNNExplainer (Ying et al., 2019), the $\lambda$ is set to default 0.05. All experiments are conducted on a Linux machine with 8 NVIDIA A100 GPUs, each with 40GB of memory. The software environment is CUDA 11.3 and Driver Version 550.54.15. We used Python 3.9.13, Pytorch 1.10.0, and torch-geometric 2.0.3 to construct our project. Detailed inforamtion are available in Appendix B.2.

## 6.2 DETECTION PERFORMANCE COMPARISON

We compare LM$^2$OTIFS against 13 baselines, including supervised and zero-shot methods, to evaluate detection performance. The summary of baselines is provided in **Appendix B.2**. We report both accuracy(ACC) and area under the receiver operating characteristic(AUC) results. Due to the deterministic LLM inference in few-show methods, we report the error bar separately. For the in-domain setting, we train and test our method on the same domain. In Table 2, we report the average results of ChatGPT-based texts detection on three datasets. LM$^2$OTIFS achieves the best performance under ACC and AUC metrics. In Table 1, we study the performance across various LLMs. As the results show, the performance is aligned with Table 2. Under the ACC metric, LM$^2$OTIFS is the best performance on average, demonstrating the ability for MGT detection. The detailed results are available in **Appendix C**. Due to the limitation of pages, we provide more experiments about cross-domain evaluation, statistical significance analysis and comparison with TextGCN in **Appendix C.1**.

Table 2: Detection comparisons on HGTs and ChatGPT-generated texts. The best and second-best results are shown in bold font and underlined. * means the model is trained on that dataset.

| Method | ACC | | | | AUC | | | |
|---|---|---|---|---|---|---|---|---|
| | HC3 | M4 | RAID | Avg. | HC3 | M4 | RAID | Avg. |
| Likelihood | 0.75 | 0.88 | 0.85 | 0.83 | **1.00** | 0.90 | 0.98 | 0.96 |
| Rank | 0.53 | 0.58 | 0.56 | 0.56 | 0.89 | 0.95 | 0.91 | 0.92 |
| LogRank | 0.70 | 0.87 | 0.84 | 0.81 | **1.00** | 0.94 | 0.97 | 0.97 |
| Entropy | 0.77 | 0.73 | 0.66 | 0.72 | 0.95 | 0.79 | 0.89 | 0.88 |
| NPR | 0.83 | 0.71 | 0.79 | 0.78 | **1.00** | 0.93 | 0.97 | 0.97 |
| LRR | 0.96 | 0.86 | 0.87 | 0.90 | **1.00** | 0.98 | 0.96 | 0.98 |
| DetectGPT | 0.63 | 0.61 | 0.62 | 0.62 | 0.56 | 0.63 | 0.78 | 0.66 |
| Fast-DetectGPT | 0.97 | 0.96 | 0.97 | 0.97 | **1.00** | 0.99 | **1.00** | 0.99 |
| DNAGPT | 0.73 | 0.68 | 0.72 | 0.71 | 0.88 | 0.86 | 0.93 | 0.89 |
| Binoculars | 0.98 | 0.94 | **0.99** | 0.97 | **1.00** | 0.98 | **1.00** | 0.99 |
| Glimpse | 0.98 | 0.94 | 0.91 | 0.94 | **1.00** | 0.98 | 0.96 | 0.98 |
| GPTZero | 0.77 | 0.75 | 0.68 | 0.73 | 0.77 | 0.75 | 0.68 | 0.73 |
| RoBERTa-QA | **1.00**\* | 0.95 | 0.80 | 0.91 | **1.00**\* | 0.99 | 0.96 | 0.98 |
| Radar | 0.66 | 0.76 | 0.77 | 0.73 | 0.52 | 0.83 | 0.95 | 0.76 |
| DeTeCtive | 0.92 | 0.93 | 0.96 | 0.93 | 0.93 | 0.94 | 0.98 | 0.95 |
| LM$^2$OTIFS | 0.97 | **0.98** | **0.99** | **0.98** | **1.00** | **1.00** | **1.00** | **1.00** |

## 6.3 EXPLANATION EVALUATION

**Quantitive Analysis**. Due to a lack of ground truth, evaluating the effectiveness of explanations remains challenging. Therefore, we follow previous work (Hooker et al., 2019; Zheng et al., 2025) using MoRF and LeRF to verify the motifs, which are popular evaluation protocols in XAI that assess the faithfulness of explanations by measuring how the model's prediction changes when the most or least relevant input attributions are sequentially removed according to explanations. For the MoRF protocol, a lower AUC indicates a more faithful explanation, whereas for LeRF, a higher AUC is better. We evaluate our motifs on the HC3 dataset using this framework. We first extract the explainable motifs, which indicate the importance of each edge. Then we remove the most important edges following an increasing sequence. As shown in Figure 3, the motifs generated by our method, LM$^2$OTIFS, are more effective and consistently outperform the baseline models. Under the MoRF protocol, removing the 20% most important edges from our motifs causes an average accuracy drop of over 15% on the HC3 dataset. In contrast, explanations from other methods result in an accuracy decline of less than 10%. Conversely, under the LeRF protocol, our motifs lead to a smaller performance drop than the baselines, demonstrating their robustness. Detailed results for each domain are provided in **Appendix C.2**.

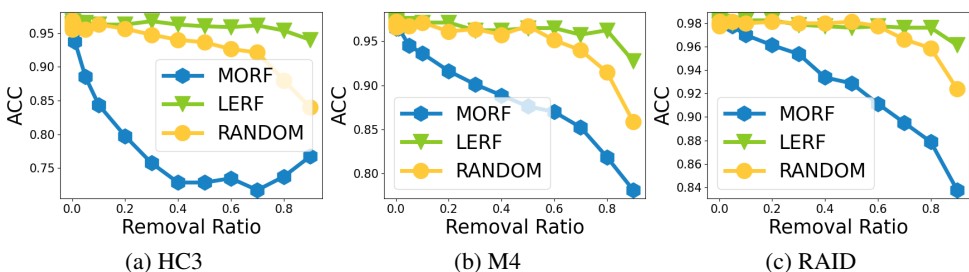

Figure 3: Comparison results of MoRF and LeRF between explainable motifs extracted from LM$^2$OTIFS and random motifs.

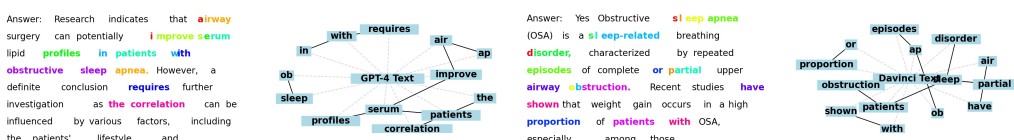

Figure 4: High-order explainable motif samples from GPT-4 and Davinci. We extract motifs from texts in the PubMed dataset for the same question. In graph motifs, solid lines represent subgraph motifs and dashed lines mean the text contains words. In text motifs, words highlighted in the same color are connected in the corresponding graph motifs. A single word may contain multiple colors.

**Qualitative Analysis.** To elucidate motif patterns, we visualize both the graph and the corresponding text motifs, encompassing word-level and higher-order structures. Word-level motifs highlight difference of word occurrence probabilities, while higher-order motifs capture complex relationships, such as phrasal and semantic structures. Figure 4 presents examples extracted from the PubMed (Jin et al., 2019) dataset, preserving the top $2\%$ of edges. While GPT-4 and Davinci share common words (e.g., "sleep", "patients"), our method captures distinct phrasal patterns. For instance, GPT-4's "ob" and "sleep" (purple) indicate "obstructive sleep", whereas Davinci's "disorder" and "sleep" represent "sleep-related breathing disorder". Furthermore, GPT-4's connection of "airway", "improves", and "serum" reveals sentence-level patterns. Detailed case studies are provided in **Appendix C.2**.

Generally, detectors make predictions by combining multiple types of features, such as word distributions and co-occurrence patterns. For example, in watermarking-based detection methods (Li et al., 2025), the probability of generating certain words is manipulated through predefined green and red lists. During inference, deviations in word frequencies can be used to determine whether the text is machine-generated. Interpretability aims to reveal which specific features contribute to the detector's decision. Importantly, these features are not merely simple statistical counts; rather, they reflect meaningful distinctions that separate different categories of text, which reveal that different language models possess *distinct and visible fingerprints*.

## 7 CONCLUSION

This paper focuses on explainable authorship detection, introducing a framework that identifies characteristic motifs to provide insight into model decisions. We evaluate our method against supervised and zero-shot learning baselines across various domains, demonstrating comparable performance. We follow the previous XAI evaluation protocol to verify the effectiveness of the explainable motifs.

**Limitation & Future Work**. First, we have not explored the impact of different GNN architectures or hyperparameter settings on the resulting explanations. Second, the quality of the explainable motifs is dependent on the quality of the graph representation, which in turn requires a sufficient number of training samples to construct effectively. Future work could investigate the robustness of our method under data-scarce conditions and explore a wider range of GNN backbones.

ETHIC STATEMENT

All authors confirm that they have read and commit to upholding the ICLR Code of Ethics. All experiments use publicly available benchmarks; no human subjects or sensitive data are involved.

REPRODUCIBILITY STATEMENT

Code is included in the Supplementary Material. All experiments use publicly available benchmarks, and are reproducible.

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

CONTENTS

## A    PROOF OF THEOREM 3.1

We first prove that the ensemble of PGB detectors is at least as accurate as the ensemble of ESB detectors. To this end, let us recall that an ESB detector is completely characterized by the mapping $g_s$ and the PGB detector by $(K, \lambda, e_h, e_m, A_t, A_s, g_p)$. Let us consider an arbitrary ESB detector by fixing the function $g_s(\cdot)$. The ESB detector computes $\widehat{P}_\ell(s_t|s_{1:t-1}), \ell \in \{h, m\}, t \in [T]$ empirically and uses $g_s((\widehat{P}_\ell(s_t|s_{1:t-1}))_{\ell \in \{h,m\}, t \in [T]}, \mathbf{S}_o)$ for detection. On the other hand, the PGB uses the embedding functions $e_\ell, A_t, A_s$ to compute the final node embeddings $\mathbf{h}^{(K)}$ and the mapping $g_p(\mathbf{h}^{(K)}, \mathbf{S}_o)$ for detection. We take $K = T$ and $\lambda = 1$. Then, to prove that there exists a PGB which matches the ESB in terms of detection accuracy, it suffices to show that there exist choices of embedding functions $e_\ell, A_t, A_s$, such that the empirical estimate $\widehat{P}_\ell(s_t|s_{1:t-1}), \ell \in \{h, m\}, t \in [T]$ can be written as a function of the final node embeddings $\mathbf{h}^{(T)}$, i.e., there exists $r(\cdot)$ such that $r(\mathbf{h}^{(T)}) = (\widehat{P}_\ell(s_t|s_{1:t-1}))_{\ell \in \{h,m\}, t \in [T]}$. Then, the proof follows by taking $g_p(r(\mathbf{h}^{(T)}), \mathbf{S}_o) = g_s((\widehat{P}_\ell(s_t|s_{1:t-1}))_{\ell \in \{h,m\}, t \in [T]}, \mathbf{S}_o)$, so that

$$P(f_{\mathrm{PGB}}(\mathcal{T}_h, \mathcal{T}_m, \mathbf{S}_o) = f_{\mathrm{ESB}}(\mathcal{T}_h, \mathcal{T}_m, \mathbf{S}_o)) = 1.$$

To this end, we take $e_\ell$ as the identity function and $A_t$ as a one-to-one parametrization function, so that for each token node $s_i$, the collection $\mathcal{J}_\ell(s_i) \times (\mathcal{I}_{\ell,j}(s_i))_{j \in \mathcal{J}_\ell(s_i)}$ can be computed from its connected edge weights, where $\mathcal{J}_\ell(s_i)$ is the training sequence indices in which the token is present and $\mathcal{I}_{\ell,j}(s_i)$ is the collection of indices in the sequence $\mathbf{S}_{\ell,j}, j \in \mathcal{J}_\ell$ whose value is equal to $s_i$. We further note that

$$\widehat{P}_\ell(s_t|s_{1:t-1}) =$$

$$\frac{1}{|\mathcal{J}_\ell(s_t)|} \sum_{i=1}^{|\mathcal{J}_\ell(s_t)|} \frac{\sum_{j=1}^{|\mathbf{S}_{\ell,i}|-t} \mathbb{1}(\mathbf{S}_{\ell,i,j:j+t} = s_{1:t})}{\sum_{j=1}^{|\mathbf{S}_{\ell,i}|-t} \mathbb{1}(\mathbf{S}_{\ell,i,j:j+t-1} = s_{1:t-1})},$$

Furthermore,

$$\mathbb{1}(\mathbf{S}_{\ell,i,j:j+t} = s_{1:t}) = \prod_{s_i: i \in [t]} \mathbb{1}((j+i) \in \mathcal{I}_{\ell,j}(s_i)),$$

Consequently, for each $t \in [T]$, the conditional distribution $\widehat{P}_\ell(s_t|s_{1:t-1})$ can be computed as a function of $\mathbf{h}^{(t)}$. As a result, the aggregate final node embedding $\mathbf{h}^{(T)}$ can yeild $\widehat{P}_\ell(s_t|s_{1:t-1}), \ell \in \{h, m\}, t \in [T]$ as a function. This complete the first part of the proof.

To prove strict improvements of PGM detectors over ESB detectors in terms of detection accuracy, we note that ESB detectors are restricted by their limited context length $T$. To provide a concrete example, consider a detection scenario characterized by the pair of probability distributions $P_h, P_m$, where all human and machine generated text sequences have length greater than $T$. That is, for any sequence $\mathbf{S}_\ell = (S_{\ell,1}, S_{\ell,2}, \cdots, S_{\ell,L})$ with $L \leq T$, we have $P_\ell(S_{\ell,1}, S_{\ell,2}, \cdots, S_{\ell,L}) = 0$, where $\ell \in \{h, m\}$. Furthermore, assume that the vocabulary consists of two tokens $\{a, b\}$. Both human and machine generated text sequences consist of tokens generated independently and with equal probability over the vocabulary for all indices in $\{1, 2, \cdots, L-1\}$. The human generated text always ends with the token $a$ and machine generated text with the token $b$, i.e., $P(S_{h,L} = a) = P(S_{m,L} = b) = 1$. Then, it is straightforward to see that a PGM can achieve accuracy equal to one, since the edge weights, which are functions of $\mathcal{J}_\ell \times (\mathcal{I}_{\ell,j})_{j \in \mathcal{J}_\ell}$ can capture the fact that the human generated text ends in $a$ and machine generated text ends in $b$. On the other hand, for an ESB, it can be noted that all of the empirical conditional distributions $\widehat{P}_\ell(s_t|s_{1:t-1}), t \in [T], \ell \in \{h, m\}$ converge to uniform Bernoulli distributions as $L \to \infty$. So, the ESB achieves an accuracy which is strictly less than 1 due to its limited context length, and its accuracy converges to $\frac{1}{2}$ as $L \to \infty$. This completes the proof. $\square$

## B    EXPERIMENTAL SETUP DETAILS

### B.1    DATASETS

Our evaluation employs six distinct datasets. We selected specific domains or text sources from each to create a comprehensive benchmark.

Table 3: The details of the dataset for detection between HGT and MGT generated by ChaGPT.

| | HC3 | | | | M4 | | | | RAID | | | |
|---|---|---|---|---|---|---|---|---|---|---|---|---|
| | open-qa | wiki-csai | medicine | finance | wiki-how | reddit | peerread | arxiv | recipe | book | poetry | review |
| # Training | 2,000 | 1,384 | 2,000 | 2,000 | 2,000 | 872 | 2,000 | 2,000 | 2,000 | 2,000 | 2,000 | 1,793 |
| # Validation | 100 | 100 | 100 | 100 | 100 | 100 | 100 | 100 | 100 | 100 | 100 | 100 |
| # Test | 200 | 200 | 200 | 200 | 200 | 200 | 200 | 200 | 200 | 200 | 200 | 200 |
| # Nodes | 15,974 | 12,069 | 8,127 | 9,581 | 20,061 | 11,276 | 18,926 | 10,526 | 6,562 | 20,515 | 16,818 | 17,024 |
| # Edges | 3,262K | 2,635K | 2,063K | 2,326K | 6,823K | 4,658K | 8,591K | 3,595K | 2,119K | 7,076K | 5,059K | 5,448K |

Table 4: The details of the M4 dataset for detection between HGT and MGT generated by LLMs.

| | reddit | | | | peerread | | | | arxiv | | | |
|---|---|---|---|---|---|---|---|---|---|---|---|---|
| | Davinci | Cohere | Dolly | BloomZ | Davinci | Cohere | Dolly | BloomZ | Davinci | Cohere | Dolly | BloomZ |
| # Training | 2,000 | 2,000 | 2,000 | 2,000 | 872 | 824 | 872 | 830 | 2,000 | 2,000 | 2,000 | 2,000 |
| # Validation | 100 | 100 | 100 | 100 | 100 | 100 | 100 | 98 | 100 | 100 | 100 | 100 |
| # Test | 200 | 200 | 200 | 200 | 200 | 198 | 200 | 192 | 200 | 200 | 200 | 200 |
| # Nodes | 20,867 | 21,701 | 21,344 | 20,944 | 11,059 | 10,837 | 14,366 | 11,340 | 10,153 | 10,724 | 12,039 | 11,468 |
| # Edges | 7,175K | 7,055K | 7,157K | 6,601K | 4,139K | 3,933K | 5,924K | 4,296K | 3,343K | 3,376K | 4,132K | 4,011K |

Table 5: The details of the RAID dataset for detection between HGT and MGT generated by LLMs.

| | recipes | | | | poetry | | | | reviews | | | |
|---|---|---|---|---|---|---|---|---|---|---|---|---|
| | Llama | GPT-4 | MPT | Mistral | Llama 2 | GPT-4 | MPT | Mistral | Llama 2 | GPT-4 | MPT | Mistral |
| # Training | 2,000 | 2,000 | 2,000 | 2,000 | 2,000 | 2,000 | 2,000 | 2,000 | 1,793 | 1,793 | 1,793 | 1,793 |
| # Validation | 100 | 100 | 100 | 100 | 100 | 100 | 100 | 100 | 100 | 100 | 100 | 100 |
| # Test | 200 | 200 | 200 | 200 | 200 | 200 | 200 | 200 | 200 | 200 | 200 | 200 |
| # Nodes | 6,904 | 6,701 | 13,466 | 8,833 | 16,696 | 17,152 | 19,476 | 17,523 | 17,004 | 17,387 | 19,371 | 17,843 |
| # Edges | 2,125K | 2,163K | 4,066K | 2,586K | 4,766K | 4,527K | 5,924K | 4,835K | 5,039K | 5,694K | 6,017K | 5,142K |

Table 6: The details of the dataset for detection between HGT and MGT generated by LLMs in Yelp, Creative, and Essay Dataset. Sonnet and Opus are short for Claude3-Sonnet and Claude-3-Opus.

| | Yelp | | | Essay | | | Creative | | |
|---|---|---|---|---|---|---|---|---|---|
| | Sonnet | Opus | Gemini | Sonnet | Opus | Gemini | Sonnet | Opus | Gemini |
| # Training | 2,000 | 2,000 | 2,000 | 1,500 | 1,500 | 1,500 | 1,500 | 1,500 | 1,500 |
| # Validation | 200 | 200 | 200 | 100 | 100 | 100 | 100 | 100 | 100 |
| # Test | 200 | 200 | 200 | 200 | 200 | 200 | 200 | 200 | 200 |
| # Nodes | 11,581 | 11,308 | 11,350 | 20,836 | 20,748 | 20,868 | 20,597 | 20,057 | 19,936 |
| # Edges | 1,940K | 1,886K | 1,778K | 8,422K | 8,038K | 7,989K | 7,187K | 6,308K | 6,250K |

- **HC3**(Human-ChatGPT Comparison Corpus) (Guo et al., 2023): This dataset contains questions with both human-generated text (HGT) and machine-generated text (MGT) from ChatGPT. From its five available domains, we utilize four for our experiments: open-qa, wiki-csai, medicine, and finance.

- **M4** (Wang et al., 2024): The M4 dataset provides MGT from several LLMs, including Davinci, Dolly, and BloomZ, across diverse domains such as wiki-how, reddit, peerread, and arxiv.

- **RAID** (Dugan et al., 2024): This large-scale dataset contains documents generated by 11 LLMs across 11 genres. Our benchmark includes four of these: recipe, book, poetry, and review.

- **Yelp, Creative, and Essay** (Mao et al., 2024; Verma et al., 2023; Guo et al., 2024a): For these three datasets, while texts from five LLMs are available, our analysis focuses on those generated by Claude-3-Sonnet, Claude-3-Opus, and Gemini-1.0-Pro.

Detailed statistics for the training, validation, and test sets, including their graph representations, are presented in Tables 3, 4, 5, and 6.

Table 7: Detection comparisons with SOTA methods on ACC between HGT and ChatGPT-generated texts. The best results are shown in bold font. The second-best results are shown underlined. * means the model is trained on that dataset. The Fast-D.GPT is short for Fast-DetectGPT.

| Method | HC3 | | | | M4 | | | | RAID | | | | Avg. |
|---|---|---|---|---|---|---|---|---|---|---|---|---|---|
| | open-qa | wiki-csai | medicine | finance | wiki-how | reddit | peerread | arxiv | recipe | book | poetry | review | |
| Likelihood | 0.85 | 0.81 | 0.76 | 0.58 | 0.85 | 0.96 | 0.80 | 0.92 | 0.83 | 0.96 | 0.82 | 0.78 | 0.83 |
| Rank | 0.54 | 0.53 | 0.54 | 0.51 | 0.57 | 0.55 | 0.58 | 0.61 | 0.51 | 0.65 | 0.53 | 0.54 | 0.56 |
| LogRank | 0.77 | 0.73 | 0.72 | 0.58 | 0.82 | 0.95 | 0.80 | 0.92 | 0.81 | 0.93 | 0.84 | 0.79 | 0.81 |
| Entropy | 0.92 | 0.76 | 0.77 | 0.61 | 0.83 | 0.81 | 0.68 | 0.60 | 0.80 | 0.71 | 0.49 | 0.62 | 0.72 |
| NPR | 0.65 | 0.92 | 0.91 | 0.85 | 0.61 | 0.68 | 0.83 | 0.72 | 0.84 | 0.83 | 0.50 | 0.97 | 0.78 |
| LRR | 0.98 | 0.95 | 0.98 | 0.94 | 0.82 | 0.82 | **1.00** | 0.81 | 0.94 | 0.81 | 0.75 | 0.98 | 0.90 |
| DetectGPT | 0.46 | 0.63 | 0.76 | 0.68 | 0.58 | 0.66 | 0.59 | 0.61 | 0.56 | 0.66 | 0.59 | 0.68 | 0.62 |
| Fast-D.GPT | 0.95 | 0.99 | 0.98 | 0.97 | 0.88 | 0.94 | **1.00** | **1.00** | 0.99 | 0.97 | 0.93 | **1.00** | 0.97 |
| DNAGPT | 0.63 | 0.79 | 0.63 | 0.88 | 0.60 | 0.79 | 0.53 | 0.81 | 0.71 | 0.82 | 0.75 | 0.59 | 0.71 |
| Binoculars | 0.92 | **1.00** | **1.00** | **1.00** | 0.77 | 0.97 | **1.00** | **1.00** | **1.00** | 0.96 | **0.99** | 0.99 | 0.97 |
| Glimpse | 0.95 | 0.98 | 0.99 | 0.99 | 0.97 | 0.91 | 0.87 | 0.99 | 0.94 | 0.96 | 0.76 | 0.98 | 0.94 |
| GPTZero | 0.58 | 0.69 | 0.96 | 0.84 | 0.54 | 0.82 | 0.96 | 0.69 | 0.61 | 0.84 | 0.48 | 0.78 | 0.73 |
| RoBERTa-QA | 1.00* | 1.00* | 1.00* | 0.99* | 0.88 | 0.96 | 0.99 | 0.95 | 0.83 | 0.86 | 0.50 | 1.00 | 0.91 |
| Radar | 0.52 | 0.81 | 0.55 | 0.75 | 0.46 | 0.93 | 0.88 | 0.77 | 0.61 | 0.97 | 0.61 | 0.89 | 0.73 |
| DeTeCtive | 0.99 | 0.79 | 0.99 | 0.89 | 0.89 | 0.93 | 0.90 | 0.98 | 0.94 | 0.95 | 0.97 | 0.97 | 0.93 |
| LM²OTIFS | 0.97 | 0.96 | 0.98 | 0.98 | **0.97** | **0.99** | 0.98 | 0.96 | 0.99 | **1.00** | **0.99** | 0.96 | **0.98** |

Table 8: MGT detection AUC performance comparisons with SOTA methods on HGT and ChatGPT-generated texts. The best results are shown in bold font. The second-best results are shown underlined. * means the model is trained on that dataset. The Fast-D.GPT is short for Fast-DetectGPT.

| Method | HC3 | | | | M4 | | | | RAID | | | | Avg. |
|---|---|---|---|---|---|---|---|---|---|---|---|---|---|
| | open-qa | wiki-csai | medicine | finance | wiki-how | reddit | peerread | arxiv | recipe | book | poetry | review | |
| Likelihood | **1.00** | **1.00** | **1.00** | **1.00** | 0.95 | 0.99 | 0.69 | 0.97 | **1.00** | **1.00** | 0.90 | **1.00** | 0.96 |
| Rank | **1.00** | 0.77 | 0.99 | 0.81 | 0.94 | 0.92 | 0.97 | 0.95 | 0.79 | 0.99 | 0.87 | **1.00** | 0.92 |
| LogRank | **1.00** | **1.00** | **1.00** | **1.00** | 0.95 | 0.99 | 0.82 | 0.98 | **1.00** | **1.00** | 0.89 | **1.00** | 0.97 |
| Entropy | 0.99 | 0.85 | 0.99 | 0.97 | 0.91 | 0.91 | 0.60 | 0.75 | 0.97 | 0.88 | 0.75 | 0.97 | 0.88 |
| NPR | **1.00** | **1.00** | **1.00** | **1.00** | 0.95 | 0.99 | 0.81 | 0.98 | **1.00** | **1.00** | 0.89 | **1.00** | 0.97 |
| LRR | **1.00** | 0.99 | **1.00** | 0.99 | 0.93 | 0.98 | **1.00** | 0.99 | 0.99 | 0.99 | 0.85 | **1.00** | 0.98 |
| DetectGPT | 0.35 | 0.59 | 0.70 | 0.61 | 0.68 | 0.71 | 0.70 | 0.43 | 0.65 | 0.77 | 0.84 | 0.84 | 0.66 |
| Fast-D.GPT | **1.00** | **1.00** | **1.00** | 0.98 | 0.96 | 0.99 | **1.00** | **1.00** | **1.00** | **1.00** | 0.98 | **1.00** | 0.99 |
| DNAGPT | 0.72 | 0.95 | 0.91 | 0.94 | 0.97 | 0.95 | 0.56 | 0.94 | 0.94 | 0.97 | 0.84 | 0.98 | 0.89 |
| Binoculars | 0.98 | **1.00** | **1.00** | **1.00** | 0.90 | **1.00** | **1.00** | **1.00** | **1.00** | **1.00** | 0.99 | **1.00** | 0.99 |
| Glimpse | **1.00** | **1.00** | **1.00** | **1.00** | **0.99** | **1.00** | 0.92 | **1.00** | **1.00** | **1.00** | 0.85 | **1.00** | 0.98 |
| GPTZero | 0.58 | 0.69 | 0.96 | 0.84 | 0.54 | 0.82 | 0.96 | 0.69 | 0.61 | 0.84 | 0.48 | 0.78 | 0.73 |
| RoBERTa-QA | 1.00* | 1.00* | 1.00* | 1.00* | 0.94 | **1.00** | **1.00** | **1.00** | 0.90 | 0.99 | 0.95 | **1.00** | 0.98 |
| Radar | 0.20 | 0.77 | 0.41 | 0.68 | 0.40 | 0.97 | **1.00** | 0.95 | 0.99 | **1.00** | 0.88 | 0.91 | 0.76 |
| DeTeCtive | **1.00** | 0.84 | 0.99 | 0.89 | 0.90 | 0.98 | 0.90 | 0.99 | 0.99 | 0.97 | 0.97 | 0.97 | 0.95 |
| LM²OTIFS | **1.00** | 0.99 | **1.00** | **1.00** | **0.99** | **1.00** | **1.00** | **1.00** | **1.00** | **1.00** | **1.00** | **1.00** | **1.00** |

## B.2 EXPERIMENTAL SETUP

**Detection Baselines**. Our evaluation includes comparisons with several zero-shot detection methods: **Likelihood, Rank, Log-Rank, Entropy** (Gehrmann et al., 2019b; Solaiman et al., 2019; Ippolito et al., 2019), **DetectGPT** (Mitchell et al., 2023), **DetectLLM** (**LRR** and **NPR**) (Su et al., 2023), **DNA-GPT** (Yang et al., 2024), **Fast-DetectGPT** (Bao et al., 2024), **Glimpse** (Bao et al., 2025) and **Binoculars** (Ma & Wang, 2024). DetectGPT employs perturbations to approximate the probability distribution of the text. Fast-DetectGPT improves upon this by introducing a conditional probability curvature metric for detector optimization, thus replacing traditional perturbation-based methods. DNA-GPT adopts a distinct approach: it first truncates the input text, then uses LLMs to generate the subsequent content, and finally analyzes the N-gram differences between the original and generated text. To make a fair comparison, we utilize the OPT-2.7B model (Zhang et al., 2022) as the default reference model. For detailed implementation specifics, we followed the publicly available implementation of Fast-DetectGPT [2].

---

[2]https://github.com/baoguangsheng/fast-detect-gpt

Table 9: Cross-domain MGT detection ACC performance comparisons with SOTA methods on HGT and ChatGPT-generated texts. The best results are shown in bold font. The second-best results are shown in underlined. The Fast-D.GPT is short for Fast-DetectGPT.

| Method | HC3 | | | M4 | | | RAID | | | Avg. |
|---|---|---|---|---|---|---|---|---|---|---|
| | wiki-csai | medicine | finance | reddit | peerread | arxiv | recipe | poetry | review | |
| Likelihood | 0.97 | 0.96 | 0.98 | 0.88 | 0.80 | 0.67 | 0.57 | 0.70 | 0.99 | 0.84 |
| Rank | 0.65 | 0.94 | 0.65 | 0.82 | 0.56 | 0.66 | 0.54 | 0.80 | 0.80 | 0.71 |
| LogRank | 0.98 | 0.97 | 0.98 | 0.92 | 0.83 | 0.71 | 0.53 | 0.73 | 0.97 | 0.85 |
| Entropy | 0.71 | 0.91 | 0.87 | 0.61 | 0.75 | 0.50 | 0.50 | 0.58 | 0.78 | 0.69 |
| NPR | 0.97 | 0.97 | 0.98 | 0.93 | 0.97 | 0.77 | 0.55 | 0.72 | 0.97 | 0.87 |
| LRR | 0.98 | 0.94 | 0.96 | 0.93 | 0.94 | 0.76 | 0.52 | 0.69 | 0.93 | 0.85 |
| DetectGPT | 0.52 | 0.58 | 0.53 | 0.56 | 0.54 | 0.50 | 0.56 | 0.67 | 0.73 | 0.58 |
| Fast-D.GPT | 0.99 | **0.99** | 0.95 | 0.93 | **1.00** | 0.94 | **1.00** | 0.89 | **1.00** | **0.97** |
| DNAGPT | 0.86 | 0.87 | 0.84 | 0.92 | 0.49 | 0.85 | 0.84 | 0.67 | 0.96 | 0.81 |
| Binoculars | 0.97 | 0.97 | 0.97 | 0.89 | 0.89 | 0.89 | **1.00** | **1.00** | **1.00** | 0.95 |
| Glimpse | **1.00** | 0.98 | **1.00** | **0.96** | 0.88 | **0.99** | 0.96 | 0.78 | **1.00** | 0.95 |
| RoBERTa-QA | 0.53 | 0.65 | 0.53 | 0.77 | 0.93 | **0.99** | 0.70 | 0.86 | 0.85 | 0.76 |
| Radar | 0.79 | 0.53 | 0.74 | 0.92 | 0.77 | 0.87 | 0.65 | 0.74 | 0.88 | 0.77 |
| DeTeCtive | 0.68 | 0.58 | 0.60 | 0.76 | 0.76 | 0.65 | 0.48 | 0.72 | 0.89 | 0.68 |
| LM$^2$OTIFS | 0.71 | 0.82 | 0.64 | 0.59 | 0.99 | 0.93 | 0.50 | 0.93 | 0.97 | 0.79 |

Table 10: MGT detection ACC performance comparisons with SOTA methods on HGT and MGT on M4 dataset. The best results are shown in bold font. The second-best results are shown in underlined. The Fast-D.GPT is short for Fast-DetectGPT.

| Method | DaVinci | | | Cohere | | | Dolly | | | BloomZ | | | Avg. |
|---|---|---|---|---|---|---|---|---|---|---|---|---|---|
| | reddit | peerread | arxiv | reddit | peerread | arxiv | reddit | peerread | arxiv | reddit | peerread | arxiv | |
| Likelihood | 0.89 | 0.77 | 0.40 | 0.95 | 0.78 | 0.89 | 0.63 | 0.65 | 0.71 | 0.56 | 0.34 | 0.72 | 0.69 |
| Rank | 0.57 | 0.51 | 0.45 | 0.56 | 0.54 | 0.52 | 0.50 | 0.53 | 0.57 | 0.55 | 0.51 | 0.53 | 0.53 |
| LogRank | 0.83 | 0.77 | 0.40 | 0.94 | 0.81 | 0.90 | 0.75 | 0.69 | 0.73 | 0.71 | 0.39 | 0.77 | 0.72 |
| Entropy | 0.78 | 0.71 | 0.37 | 0.73 | 0.53 | 0.58 | 0.54 | 0.46 | 0.58 | 0.63 | 0.34 | 0.62 | 0.57 |
| NPR | 0.67 | 0.74 | 0.49 | 0.65 | 0.83 | 0.53 | 0.51 | 0.52 | 0.61 | 0.52 | 0.71 | 0.55 | 0.61 |
| LRR | 0.86 | 0.95 | 0.50 | 0.68 | 0.94 | 0.63 | 0.75 | 0.82 | 0.66 | 0.75 | 0.98 | 0.59 | 0.76 |
| DetectGPT | 0.56 | 0.53 | 0.35 | 0.63 | 0.60 | 0.47 | 0.54 | 0.46 | 0.45 | 0.58 | 0.57 | 0.62 | 0.53 |
| Fast-D.GPT | 0.97 | **1.00** | 0.46 | 0.96 | 0.99 | **0.98** | 0.90 | 0.99 | 0.82 | 0.43 | 0.51 | 0.69 | 0.81 |
| DNAGPT | 0.75 | 0.47 | 0.36 | 0.90 | 0.47 | 0.86 | 0.51 | 0.53 | 0.54 | 0.45 | 0.49 | 0.57 | 0.58 |
| Binoculars | **0.98** | **1.00** | 0.51 | **0.98** | 0.96 | **0.98** | 0.83 | 0.99 | 0.87 | 0.58 | 0.62 | 0.77 | 0.84 |
| Glimpse | 0.77 | 0.95 | 0.51 | 0.95 | 0.88 | 1.00 | 0.64 | 0.68 | 0.75 | 0.52 | 0.43 | 0.88 | 0.75 |
| GPTZero | 0.86 | 0.99 | 0.36 | 0.84 | 0.92 | 0.65 | 0.76 | 0.58 | 0.50 | 0.61 | 0.53 | 0.46 | 0.67 |
| RoBERTa-QA | 0.93 | 1.00 | 0.55 | 0.95 | 0.97 | 0.89 | 0.95 | 0.55 | 0.71 | 0.50 | 0.50 | 0.52 | 0.75 |
| Radar | 0.84 | 0.88 | 0.57 | 0.87 | 0.85 | 0.60 | 0.66 | 0.77 | 0.53 | 0.80 | 0.79 | 0.30 | 0.71 |
| DeTeCtive | 0.90 | 0.85 | **0.95** | 0.84 | 0.76 | 0.95 | **0.96** | 0.75 | **0.98** | 0.94 | 0.89 | 0.92 | 0.89 |
| LM$^2$OTIFS | 0.97 | **1.00** | 0.87 | **0.98** | **1.00** | 0.94 | 0.95 | **1.00** | 0.77 | **1.00** | **1.00** | **0.95** | **0.95** |

Our comparative evaluation also includes training-based methods: **RoBERTa-QA** (Guo et al., 2023), **DeTeCtive** (Guo et al., 2024b), and **RADAR** (Hu et al., 2023). Additionally, we present comparison results with **GPTZero** [3]. DeTeCtive is specifically designed for multi-source MGT detection. It employs contrastive learning to minimize the representational divergence among various MGT sources. During prediction, DeTeCtive utilizes k-nearest neighbors (KNN) to determine the classification. For our experiments, we use the DeTeCtive model trained on the OUTFOX dataset (Koike et al., 2024). RoBERTa-QA, proposed in (Guo et al., 2023) and trained on the HC3 dataset, leverages the pre-trained RoBERTa model (Liu, 2019) and fine-tunes a classification layer on the HC3 data.

---

[3]https://gptzero.me

Table 11: MGT detection ACC performance comparisons with SOTA methods on HGT and MGT on RAID dataset. The best results are shown in bold font. The second-best results are shown in underlined. The Fast-D.GPT is short for Fast-DetectGPT.

| Method | Llama | | | GPT-4 | | | MPT | | | Mistral | | | Avg. |
|---|---|---|---|---|---|---|---|---|---|---|---|---|---|
| | recipe | poetry | review | recipe | poetry | review | recipe | poetry | review | recipe | poetry | review | |
| Likelihood | 0.83 | 0.78 | 0.76 | 0.82 | 0.68 | 0.75 | 0.27 | 0.69 | 0.54 | 0.45 | 0.76 | 0.73 | 0.67 |
| Rank | 0.51 | 0.53 | 0.54 | 0.51 | 0.53 | 0.54 | 0.50 | 0.53 | 0.50 | 0.50 | 0.53 | 0.54 | 0.52 |
| LogRank | 0.79 | 0.81 | 0.79 | 0.80 | 0.64 | 0.78 | 0.30 | 0.63 | 0.44 | 0.43 | 0.77 | 0.78 | 0.66 |
| Entropy | 0.78 | 0.47 | 0.61 | 0.76 | 0.49 | 0.60 | 0.29 | 0.65 | 0.62 | 0.55 | 0.68 | 0.65 | 0.60 |
| NPR | 0.92 | 0.50 | 0.94 | 0.73 | 0.50 | 0.76 | 0.56 | 0.53 | 0.54 | 0.58 | 0.53 | 0.84 | 0.66 |
| LRR | 0.91 | 0.81 | 0.90 | 0.78 | 0.60 | 0.74 | 0.57 | 0.53 | 0.56 | 0.66 | 0.61 | 0.86 | 0.71 |
| DetectGPT | 0.51 | 0.77 | 0.73 | 0.51 | 0.61 | 0.65 | 0.44 | 0.48 | 0.46 | 0.51 | 0.52 | 0.56 | 0.56 |
| Fast-D.GPT | 0.92 | 0.94 | 0.96 | 0.96 | 0.79 | 0.80 | 0.39 | 0.63 | 0.41 | 0.48 | 0.79 | 0.64 | 0.73 |
| DNAGPT | 0.76 | 0.69 | 0.58 | 0.68 | 0.70 | 0.59 | 0.29 | 0.54 | 0.35 | 0.40 | 0.52 | 0.70 | 0.57 |
| Binoculars | **1.00** | **0.98** | 0.95 | **0.99** | 0.81 | 0.95 | 0.43 | 0.62 | 0.68 | 0.76 | 0.72 | 0.65 | 0.80 |
| Glimpse | 0.93 | 0.77 | 0.95 | 0.93 | 0.60 | 0.79 | 0.70 | 0.59 | 0.75 | 0.81 | 0.67 | 0.84 | 0.78 |
| GPTZero | 0.74 | 0.47 | 0.73 | 0.61 | 0.46 | 0.73 | 0.53 | 0.52 | 0.57 | 0.58 | 0.57 | 0.51 | 0.59 |
| RoBERTa-QA | 0.85 | 0.50 | 0.96 | 0.76 | 0.50 | 0.83 | 0.46 | 0.53 | 0.69 | 0.44 | 0.55 | 0.69 | 0.65 |
| Radar | 0.58 | 0.59 | 0.86 | 0.63 | 0.57 | 0.86 | 0.59 | 0.73 | 0.59 | 0.64 | **0.89** | 0.63 | 0.68 |
| DeTeCtive | **1.00** | 0.95 | 0.94 | 0.97 | 0.96 | 0.97 | 0.91 | **0.90** | **0.95** | 0.87 | 0.88 | 0.90 | 0.93 |
| LM²OTIFS | **1.00** | **0.98** | **0.97** | **0.99** | **1.00** | **1.00** | **0.95** | 0.84 | 0.90 | **0.94** | 0.88 | **0.92** | **0.95** |

Table 12: MGT detection ACC performance comparisons with SOTA methods on HGT and MGT on Yelp, Essay, and Creative dataset. The best results are shown in bold font. The second-best results are shown in underlined. The Fast-D.GPT is short for Fast-DetectGPT.

| Method | Claude3-Sonnet | | | Claude3-Opus | | | Gemini | | | Avg. |
|---|---|---|---|---|---|---|---|---|---|---|
| | Yelp | Essay | Creative | Yelp | Essay | Creative | Yelp | Essay | Creative | |
| Likelihood | 0.61 | 0.96 | 0.83 | 0.61 | 0.97 | 0.91 | 0.56 | 0.97 | 0.69 | 0.79 |
| Rank | 0.51 | 0.54 | 0.51 | 0.50 | 0.55 | 0.51 | 0.50 | 0.55 | 0.51 | 0.52 |
| LogRank | 0.57 | 0.91 | 0.79 | 0.54 | 0.93 | 0.89 | 0.54 | 0.94 | 0.68 | 0.75 |
| Entropy | 0.60 | 0.87 | 0.69 | 0.58 | 0.92 | 0.71 | 0.53 | 0.85 | 0.53 | 0.70 |
| NPR | 0.62 | 0.67 | 0.80 | 0.62 | 0.58 | 0.68 | 0.50 | 0.57 | 0.56 | 0.62 |
| LRR | 0.55 | 0.90 | 0.78 | 0.52 | 0.91 | 0.73 | 0.45 | 0.58 | 0.56 | 0.66 |
| DetectGPT | 0.49 | 0.68 | 0.69 | 0.44 | 0.62 | 0.69 | 0.42 | 0.66 | 0.62 | 0.59 |
| Fast-D.GPT | 0.66 | **1.00** | 0.88 | 0.72 | 0.99 | 0.93 | 0.60 | **0.98** | 0.69 | 0.83 |
| DNAGPT | 0.54 | 0.66 | 0.66 | 0.54 | 0.71 | 0.67 | 0.53 | 0.77 | 0.64 | 0.64 |
| Binoculars | 0.69 | **1.00** | 0.94 | 0.77 | **1.00** | 0.97 | 0.68 | 0.97 | **0.78** | 0.87 |
| Glimpse | 0.69 | **1.00** | 0.86 | 0.69 | 0.97 | 0.90 | 0.59 | 0.96 | 0.74 | 0.82 |
| GPTZero | 0.63 | 0.66 | 0.78 | 0.61 | 0.65 | 0.86 | 0.59 | 0.36 | 0.66 | 0.64 |
| RoBERTa-QA | 0.72 | 0.86 | 0.79 | 0.82 | 0.87 | 0.93 | 0.81 | 0.86 | 0.72 | 0.82 |
| Radar | 0.62 | 0.94 | 0.84 | 0.64 | 0.95 | 0.91 | 0.64 | 0.96 | 0.74 | 0.80 |
| DeTeCtive | 0.98 | 0.86 | 0.97 | 0.99 | 0.79 | 0.96 | 0.97 | 0.85 | 0.77 | 0.90 |
| LM²OTIFS | **0.99** | 0.99 | **0.98** | **1.00** | 0.99 | **0.98** | **0.99** | 0.97 | 0.77 | **0.96** |

**Explainable Baselines**. To verify the effectiveness of our method, we introduce a simple baseline, Random Motifs, which serves as a graph-explainable sanity check, where the importance of each edge is randomly assigned. If an explanation method performs worse than random, it is considered to provide no meaningful insight.

**Implementation.** The detector is implemented as a two-layer Graph Convolutional Network. The input dimension of the first layer is dependent on the token size of the training set. The hidden dimension is 64, and the output dimensionality is fixed to the number of text categories. We use the Bert (Devlin et al., 2019) tokenizer as the tokenizer. We employ Adam (Kingma, 2014) as the default optimizer with the learning rate 5E-4, 5000 epochs. For motif extraction, we adapt the GNNExplainer (Ying et al., 2019) to suit our analysis. Notably, the explanation method can be replaced by others, and we only use a basic post-hoc explainer here. We follow the Refine (Wang

Table 13: MGT detection AUC performance comparisons with SOTA methods on HGT and MGT on M4 dataset. The best results are shown in bold font. The second-best results are shown in underlined. The Fast-D.GPT is short for Fast-DetectGPT.

| Method | DaVinci | | | Cohere | | | Dolly | | | BloomZ | | | Avg. |
|---|---|---|---|---|---|---|---|---|---|---|---|---|---|
| | reddit | peerread | arxiv | reddit | peerread | arxiv | reddit | peerread | arxiv | reddit | peerread | arxiv | |
| Likelihood | 0.98 | 0.83 | 0.27 | 0.96 | 0.78 | 0.96 | 0.93 | 0.60 | 0.80 | 0.70 | 0.47 | 0.78 | 0.76 |
| Rank | 0.92 | 0.94 | 0.45 | 0.90 | 0.82 | 0.81 | 0.72 | 0.50 | 0.69 | 0.88 | 0.72 | 0.88 | 0.77 |
| LogRank | 0.98 | 0.96 | 0.28 | 0.97 | 0.90 | 0.97 | 0.93 | 0.65 | 0.79 | 0.84 | 0.58 | 0.85 | 0.81 |
| Entropy | 0.86 | 0.58 | 0.23 | 0.76 | 0.61 | 0.58 | 0.76 | 0.51 | 0.61 | 0.83 | 0.49 | 0.69 | 0.63 |
| NPR | 0.98 | **1.00** | 0.28 | 0.97 | **1.00** | 0.97 | 0.86 | 0.97 | 0.80 | 0.86 | 0.97 | 0.86 | 0.88 |
| LRR | 0.97 | **1.00** | 0.36 | 0.97 | **1.00** | 0.97 | 0.98 | 0.92 | 0.74 | _0.98_ | **1.00** | _0.93_ | 0.90 |
| DetectGPT | 0.59 | 0.74 | 0.29 | 0.72 | 0.76 | 0.43 | 0.61 | 0.54 | 0.42 | 0.73 | 0.71 | 0.63 | 0.60 |
| Fast-D.GPT | _0.99_ | **1.00** | 0.48 | _0.99_ | **1.00** | _0.99_ | 0.97 | **1.00** | 0.90 | 0.37 | 0.52 | 0.75 | 0.83 |
| DNAGPT | 0.84 | 0.27 | 0.32 | 0.94 | 0.35 | 0.93 | 0.72 | 0.55 | 0.70 | 0.47 | 0.11 | 0.69 | 0.57 |
| Binoculars | **1.00** | **1.00** | 0.51 | 0.98 | **1.00** | **1.00** | _0.98_ | **1.00** | 0.95 | 0.53 | 0.66 | 0.85 | 0.87 |
| Glimpse | 0.92 | **1.00** | 0.51 | 0.98 | 0.96 | **1.00** | 0.83 | 0.81 | 0.91 | 0.66 | 0.42 | **0.98** | 0.83 |
| GPTZero | 0.86 | 0.99 | 0.36 | 0.84 | _0.92_ | 0.65 | 0.76 | 0.58 | 0.50 | 0.61 | 0.53 | 0.46 | 0.67 |
| RoBERTa-QA | _0.99_ | **1.00** | 0.94 | _0.99_ | **1.00** | **1.00** | _0.98_ | _0.95_ | _0.99_ | 0.61 | 0.38 | 0.66 | 0.87 |
| Radar | 0.95 | **1.00** | 0.48 | 0.97 | **1.00** | 0.78 | 0.79 | 0.92 | 0.43 | 0.90 | 0.88 | 0.52 | 0.80 |
| DeTeCtive | 0.96 | 0.85 | **0.98** | 0.89 | 0.88 | 0.98 | 0.96 | 0.86 | **1.00** | 0.96 | _0.96_ | **0.98** | _0.94_ |
| LM$^2$OTIFS | _0.99_ | **1.00** | _0.94_ | **1.00** | **1.00** | 0.98 | **0.99** | **1.00** | 0.85 | **1.00** | **1.00** | **0.98** | **0.98** |

Table 14: MGT detection AUC performance comparisons with SOTA methods on HGT and MGT on RAID dataset. The best results are shown in bold font. The second-best results are shown in underlined. The Fast-D.GPT is short for Fast-DetectGPT.

| Method | Llama | | | GPT-4 | | | MPT | | | Mistral | | | Avg. |
|---|---|---|---|---|---|---|---|---|---|---|---|---|---|
| | recipe | poetry | review | recipe | poetry | review | recipe | poetry | review | recipe | poetry | review | |
| Likelihood | _0.99_ | 0.86 | _0.98_ | 0.98 | 0.72 | 0.95 | 0.38 | 0.67 | 0.53 | 0.64 | 0.80 | 0.71 | 0.77 |
| Rank | 0.88 | 0.79 | 0.97 | 0.67 | 0.65 | 0.87 | 0.45 | 0.92 | 0.83 | 0.45 | 0.93 | 0.90 | 0.78 |
| LogRank | 0.99 | 0.87 | _0.98_ | 0.97 | 0.69 | 0.94 | 0.38 | 0.73 | 0.60 | 0.64 | 0.81 | 0.74 | 0.78 |
| Entropy | 0.94 | 0.63 | 0.92 | 0.91 | 0.59 | 0.77 | 0.35 | 0.72 | 0.61 | 0.59 | 0.80 | 0.71 | 0.71 |
| NPR | _0.99_ | 0.87 | _0.98_ | 0.97 | 0.70 | 0.93 | 0.39 | 0.74 | 0.61 | 0.64 | 0.82 | 0.74 | 0.78 |
| LRR | 0.98 | 0.88 | _0.98_ | 0.94 | 0.60 | 0.83 | 0.44 | 0.83 | 0.84 | 0.62 | 0.89 | 0.84 | 0.81 |
| DetectGPT | 0.52 | 0.82 | 0.83 | 0.55 | 0.63 | 0.75 | 0.29 | 0.45 | 0.45 | 0.48 | 0.45 | 0.55 | 0.56 |
| Fast-D.GPT | _0.99_ | _0.97_ | 0.97 | _0.99_ | 0.88 | _0.99_ | 0.50 | 0.61 | 0.51 | 0.70 | 0.77 | 0.65 | 0.79 |
| DNAGPT | 0.96 | 0.75 | 0.95 | 0.80 | 0.75 | 0.88 | 0.38 | 0.55 | 0.49 | 0.57 | 0.72 | 0.60 | 0.70 |
| Binoculars | _0.99_ | _0.99_ | 0.97 | **1.00** | _0.98_ | **0.99** | 0.55 | 0.66 | 0.59 | 0.72 | 0.79 | 0.68 | 0.83 |
| Glimpse | **1.00** | 0.87 | 0.97 | _0.99_ | 0.60 | 0.88 | 0.69 | 0.63 | 0.84 | 0.86 | 0.74 | 0.92 | 0.83 |
| GPTZero | 0.74 | 0.47 | 0.73 | 0.61 | 0.46 | 0.73 | 0.53 | 0.52 | 0.57 | 0.58 | 0.57 | 0.51 | 0.59 |
| RoBERTa-QA | 0.95 | 0.94 | 0.96 | 0.82 | 0.83 | 0.95 | 0.45 | 0.73 | 0.63 | 0.31 | 0.65 | 0.54 | 0.73 |
| Radar | 0.98 | 0.85 | 0.89 | _0.99_ | 0.81 | 0.87 | _0.95_ | 0.83 | 0.74 | 0.80 | 0.86 | 0.63 | 0.85 |
| DeTeCtive | **1.00** | 0.95 | 0.96 | _0.99_ | 0.96 | _0.99_ | 0.92 | **0.91** | **0.99** | _0.91_ | _0.90_ | _0.97_ | _0.95_ |
| LM$^2$OTIFS | **1.00** | **1.00** | **1.00** | **1.00** | **1.00** | **1.00** | **0.99** | _0.90_ | _0.95_ | **0.99** | **0.94** | **0.98** | **0.98** |

et al., 2021a) to implement the GNNExplainer. The optimizer for GNNExplainer is Adam with a learning rate of 1E-3, 100 epochs.

## C    DETAILED EXPERIMENT RESULTS

### C.1    EXTENDED DETECTION EXPERIMENTS

In our experiments, we consider in-domain detection and cross-domain detection in the same dataset and report the results in this section. We report the results under ACC and AUC metrics. For GPTZero, since it provides a binary output, we consider its ACC and AUC values to be equivalent.

**In-Domain Detection**. We provide the detailed experiment results for distinguishing HGTs and MGTs by ChatGPT in Table 7 and Table 8. The results demonstrate that LM$^2$OTIFS achieves the best performance across all domains under both ACC and AUC metrics, aligned with our analysis. In

Table 15: MGT detection ACC performance comparisons with SOTA methods on HGT and MGT on Yelp, Essay, and Creative dataset. The best results are shown in bold font. The second-best results are shown in underlined. The Fast-D.GPT is short for Fast-DetectGPT.

| Method | Claude3-Sonnet | | | Claude3-Opus | | | Gemini | | | Avg. |
|---|---|---|---|---|---|---|---|---|---|---|
| | Yelp | Essay | Creative | Yelp | Essay | Creative | Yelp | Essay | Creative | |
| Likelihood | 0.73 | 0.94 | 0.94 | 0.72 | **1.00** | **0.99** | 0.55 | **0.99** | 0.76 | 0.85 |
| Rank | 0.54 | 0.85 | 0.85 | 0.49 | 0.99 | 0.92 | 0.39 | 0.97 | 0.65 | 0.74 |
| LogRank | 0.69 | 0.93 | 0.93 | 0.68 | **1.00** | 0.98 | 0.50 | **0.99** | 0.74 | 0.83 |
| Entropy | 0.64 | 0.83 | 0.83 | 0.57 | 0.95 | 0.88 | 0.42 | 0.91 | 0.57 | 0.73 |
| NPR | 0.68 | 0.99 | 0.94 | 0.66 | 0.99 | 0.98 | 0.49 | 0.98 | 0.76 | 0.83 |
| LRR | 0.54 | **1.00** | 0.88 | 0.52 | **1.00** | 0.95 | 0.39 | **0.99** | 0.70 | 0.77 |
| DetectGPT | 0.53 | 0.75 | 0.71 | 0.43 | 0.74 | 0.78 | 0.37 | 0.80 | 0.64 | 0.64 |
| Fast-D.GPT | 0.73 | **1.00** | 0.94 | 0.81 | **1.00** | **0.99** | 0.68 | **0.99** | **0.79** | 0.88 |
| DNAGPT | 0.67 | 0.94 | 0.86 | 0.70 | 0.94 | 0.93 | 0.58 | 0.95 | 0.75 | 0.81 |
| Binoculars | 0.79 | **1.00** | 0.99 | 0.87 | **1.00** | **1.00** | 0.73 | 0.99 | 0.79 | 0.91 |
| Glimpse | 0.78 | **1.00** | 0.90 | 0.83 | **1.00** | 0.96 | 0.74 | **1.00** | 0.78 | 0.89 |
| GPTZero | 0.63 | 0.66 | 0.78 | 0.61 | 0.65 | 0.86 | 0.59 | 0.36 | 0.66 | 0.64 |
| RoBERTa-QA | 0.92 | 0.95 | 0.94 | 0.96 | 0.98 | 0.97 | 0.96 | 0.94 | 0.78 | 0.93 |
| Radar | 0.58 | 0.93 | 0.99 | 0.68 | 0.99 | 0.97 | 0.70 | **0.99** | 0.76 | 0.84 |
| DeTeCtive | 0.98 | 0.86 | 0.96 | 0.99 | 0.79 | **0.99** | 0.99 | 0.85 | 0.76 | 0.91 |
| LM$^2$OTIFS | **1.00** | **1.00** | **1.00** | **1.00** | **1.00** | 0.99 | **1.00** | **0.99** | 0.78 | **0.97** |

addition, we also provide the experiment results between HGT and MGT by other LLMs in Table 10, 11, 12, 13, 14, and 15. LM$^2$OTIFS performs consistently well on various LLMs and achieves the best performance, indicating the effectiveness of PGM for MGT detection tasks.

Table 16: Statistical significance analysis on HC3 dataset. We repeat the experiments 5 times and report the mean and standard deviation.

| Metric | open-qa | wiki-csai | medicine | finance |
|---|---|---|---|---|
| ACC | $0.9690_{\pm 0.0073}$ | $0.9410_{\pm 0.0097}$ | $0.9750_{\pm 0.0032}$ | $0.9810_{\pm 0.0037}$ |
| AUC | $0.9965_{\pm 0.0005}$ | $0.9938_{\pm 0.0005}$ | $0.9993_{\pm 0.0001}$ | $0.9983_{\pm 0.0004}$ |

**Cross-Domain Detection**. To further analysis the generality of LM$^2$OTIFS, we conduct cross-domain detection experiments. We use the open-qa, wiki-how, and books domains in HC, M4, and RAID datasets as the training domain and test on other domains, respectively. For the zero-shot baselines and RADAR, we use the training data as a reference to learning a threshold and apply it to the test domain. For the RoBERTa-QA, we follow its pipeline to fine-tune the RoBERTa on one domain and test on other domains. As Table 9 shows, LM$^2$OTIFS performs poorly on some domains, such as the reddit domain on the M4 dataset. One potential reason is that our method is only trained on a limited training set and lacks generalization, while other methods, such as zero-shot methods, fully utilize the generalization of LLM.

Table 17: MGT detection performance comparison on HC3 dataset between default(Bert) and GPT2 tokenizers.

| Metric | open-qa | | wiki-csai | | medicine | | finance | |
|---|---|---|---|---|---|---|---|---|
| | Bert | GPT2 | Bert | GPT2 | Bert | GPT2 | Bert | GPT2 |
| ACC | 0.97 | 0.99 | 0.96 | 0.95 | 0.98 | 0.98 | 0.98 | 0.97 |
| AUC | 1.00 | 1.00 | 0.99 | 1.00 | 1.00 | 0.99 | 1.00 | 1.00 |

**Statistical Significance Analysis**. To further demonstrate the robustness of LM$^2$OTIFS, we conducted a Statistical Significance Analysis. Specifically, we repeated our experiments five times on the HC3 dataset, each with a distinct random seed, and the resulting performance metrics are detailed in

Table 18: Ablation analysis on HC3 dataset. The best results are shown in bold font.

| | Method | open-qa | wiki-csai | medicine | finance | Avg. |
|---|---|---|---|---|---|---|
| ACC | LM$^2$OTIFS | 0.97 | 0.96 | 0.98 | **0.98** | **0.97** |
| | LM$^2$OTIFS-U | 0.95 | 0.94 | **1.00** | **0.98** | **0.97** |
| | LM$^2$OTIFS-W | **1.00** | 0.84 | **1.00** | 0.94 | 0.95 |
| | LM$^2$OTIFS-UW | 0.98 | 0.79 | **1.00** | 0.93 | 0.92 |
| | LM$^2$OTIFS-Bert | **1.00** | 0.79 | 0.98 | 0.89 | 0.91 |
| AUC | LM$^2$OTIFS | **1.00** | 0.99 | **1.00** | **1.00** | **1.00** |
| | LM$^2$OTIFS-U | 0.99 | **1.00** | **1.00** | **1.00** | **1.00** |
| | LM$^2$OTIFS-W | **1.00** | 0.84 | **1.00** | 0.98 | 0.96 |
| | LM$^2$OTIFS-UW | **1.00** | 0.86 | **1.00** | 0.97 | 0.96 |
| | LM$^2$OTIFS-Bert | **1.00** | 0.85 | 0.99 | 0.97 | 0.95 |

Table 19: Sliding window size ablation analysis on HC3 dataset. The best results are shown in bold font.

| | Method | open-qa | wiki-csai | medicine | finance | Avg. |
|---|---|---|---|---|---|---|
| ACC | 10 | 0.95 | 0.93 | 0.99 | 0.93 | 0.95 |
| | 15 | 0.97 | 0.95 | 0.99 | 0.94 | 0.96 |
| | 20 | 0.97 | **0.96** | 0.98 | **0.98** | **0.97** |
| | 25 | 0.96 | 0.94 | **1.00** | 0.94 | 0.96 |
| | 30 | **0.98** | 0.93 | 0.99 | 0.93 | 0.96 |
| AUC | 10 | 0.99 | 0.99 | **1.00** | 0.98 | 0.99 |
| | 15 | **1.00** | 0.99 | **1.00** | 0.98 | 0.99 |
| | 20 | **1.00** | 0.99 | **1.00** | **1.00** | **1.00** |
| | 25 | **1.00** | **1.00** | **1.00** | 0.98 | **1.00** |
| | 30 | **1.00** | **1.00** | **1.00** | 0.98 | **1.00** |

Table 16. The consistently high performance across these different runs indicates the stable and reliable nature of LM$^2$OTIFS.

**Ablation Study**. To investigate the impact of different graph characteristics on the MGT detection task, we performed ablation experiments on graph categories, specifically comparing undirected versus directed graphs and weighted versus unweighted graphs. To verify the influence of token semantics on detection performance, we also performed an ablation study on the token node initialization method, where token nodes are initialized using Bert token embeddings. In our experiments, we use -U and -W to represent undirected graphs and weighted graphs, while -Bert indicates replacing Bert token embeddings with a simpler initialization method.

We further investigated the impact of different tokenizers on the MGT detection task. Our default tokenizer is Bert's tokenizer. To assess the influence of tokenization, we conducted experiments using GPT-2's tokenizer. The results of this comparison are presented in Table 17. Our findings indicate that the choice between Bert's and GPT-2's tokenizers did not significantly affect the overall detection performance.

To investigate the effect of sliding window size on detection performance, we conduct ablation studies, and the results are presented in Table 19. As the window size increases, the detection accuracy initially improves and then declines. Based on these results, we set 20 as the default sliding-window size in our experiments.

**Time Consumption**. Compared to other training-based methods, LM$^2$OTIFS have an additional pipeline, the graph construction phase. Specifically, its time complexity for graph construction is $O(LW^2)$, where $L$ represents the length of the sentence and $W$ denotes the size of the sliding window. We also evaluated the test time efficiency of LM$^2$OTIFS in comparison to several other baselines. As detailed in Table 20, LM$^2$OTIFS demonstrates the lowest time consumption during the testing phase.

Table 20: Inference time(seconds) comparison on HC3 dataset. We repeat the experiments 10 times and report the average time consumption. - indicates the inference time is more than 10 minutes. The best results are shown in bold font.

| | open-qa | wiki-csai | medicine | finance |
|---|---|---|---|---|
| NPR | - | - | - | - |
| DNA-GPT | - | - | - | - |
| DetectGPT | 442.0000 | 161.6530 | 82.2744 | 255.4350 |
| Fast-DetectGPT | 28.0217 | 27.0673 | 24.1440 | 28.4082 |
| RoBERTa-QA | 2.9267 | 2.5464 | 2.5391 | 2.5413 |
| DeTeCtive | 18.7223 | 13.2559 | 17.2883 | 17.5105 |
| LM$^2$OTIFS | **0.0091** | **0.0065** | **0.0051** | **0.0058** |

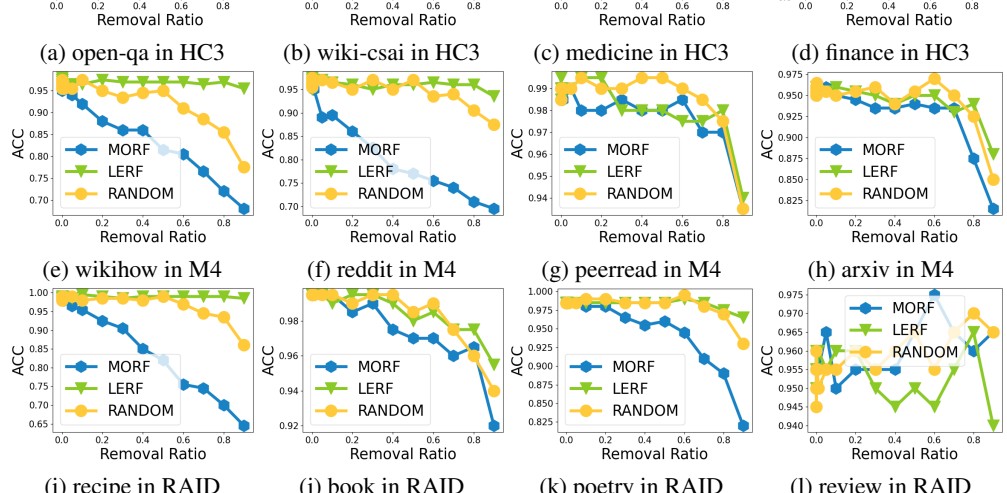

Figure 5: Comparison results of MORF and LERF between explainable motifs extracted from LM$^2$OTIFS and random motifs on HGT and ChatGPT-generated texts.

## C.2 EXTENDED MOTIFS EVALUATION

**XAI Protocol Evaluation**. We follow Section 6.3 to report the explainable motifs evaluation results on the HGT and ChatGPT-generated datasets. As the detailed results show in Figure 5, the explainable motifs are effective in most cases and obtain better results than baselines from both LeRF and MoRF protocols. However, in the medicine domain in HC3, the explainable motifs are not better than random motifs. The potential reason could be the distributed nature of the explainable motifs across numerous nodes and edges. Consequently, the deletion of some edges does not drastically impede the graph network's ability to accurately perform detection. For instance, in the medicine domain of the HC3 dataset, a significant performance drop in the GNN is observed when the proportion of deleted edges surpasses 70%.

**Extensive Evaluation**. Although interpretable approaches for the HGT detection task are currently limited, we adapt several existing interpretability methods to this task in order to demonstrate the effectiveness of our approach. Beside **random motifs**, we compare it against other baselines: **LIME** (Ribeiro et al., 2016), **SHAP** (Lundberg & Lee, 2017), **GLTR** (Gehrmann et al., 2019a), and **GPT-4o** (OpenAI, 2024). For these baselines, we use RoBERTa-QA, a well-trained model in the HC3 dataset, as the model to be explained. Notably, the reason we use RoBERTa-QA is that it has

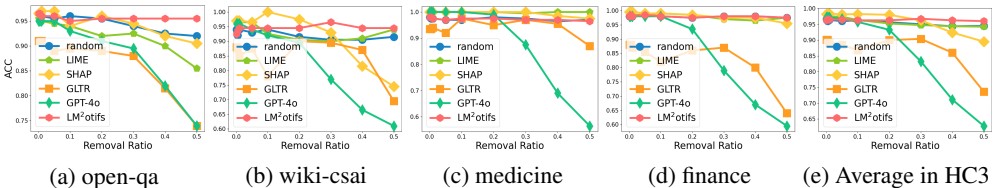

|  (a) open-qa  |  (b) wiki-csai  |  (c) medicine  |  (d) finance  |  (e) Average in HC3  |

Figure 6: Comparison results of LERF between LM$^2$OTIFS and adapted baselines.

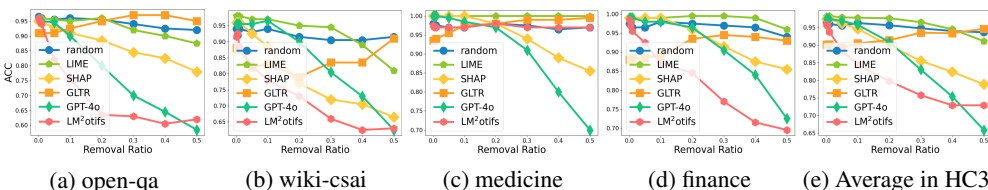

|  (a) open-qa  |  (b) wiki-csai  |  (c) medicine  |  (d) finance  |  (e) Average in HC3  |

Figure 7: Comparison results of MORF between LM$^2$OTIFS and adapted baselines.

```
prompt = ( "Text: " + text + "\n\n" + f"A pretrained Roberta Model Prediction Score: {prediction_score:.4f} (>0.5 indicates AI-generated)\n\n" +
    "Please analyze the text and provide a list of words with their importance scores (0-1).\n\n" + "Format your response EXACTLY like this
example:\n" + '[\n' + ' {"words": ["This","is","a","apple"], "score": [0.85,0.78,0.75,0.72]}\n' + ']\n\n' + "Provide all the words in the text,for each
word:\n" + "- Score should be between 0 and 1\n" + "- Higher scores (closer to 1) indicate stronger evidence of AI generation\n\n" + "Return ONLY
the JSON list, no other text."   )
```

Figure 8: The prompt of GPT-4o as an explainer.

the best performance in the HC3 dataset, and it can be replaced by other models. GLTR is a tool to analyze a piece of text and visualizing these statistical patterns, which uses a language model to determine the probability of each word appearing in its context. In this paper, we follow the original code to use GPT2 as the default language model. Besides, we also consider using the GPT-4o as a baseline for the LLM explainer. The prompt is shown in Figure 8.

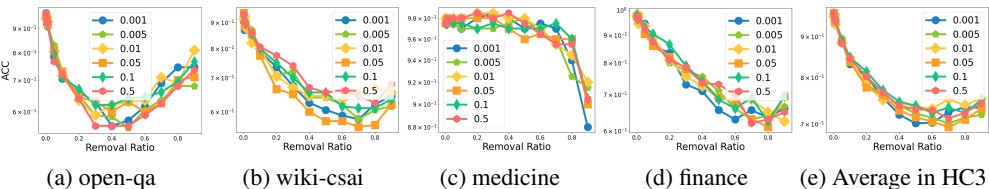

|  (a) open-qa  |  (b) wiki-csai  |  (c) medicine  |  (d) finance  |  (e) Average in HC3  |

Figure 9: Comparison results of MORF between different $\lambda$ settings on HC3 dataset.

Our results are provided in Figure 6 and 7. As Figure 3 shows, the motifs from LM$^2$OTIFS are more effective, which consistently have a better performance than baselines. From the MoRF protocol, when the 20% important edges are removed, the explainable motifs cause more than an average 15% accuracy drop on HC3 dataset, while other explanations get less than 10% accuracy decline. Under the LeRF protocol, the explainable motifs cause a lower performance drop than other motifs. GPT-4o performs well under the MoRF setting but fails under the LeRF setting.

Table 21: MORF average results of the $\lambda$ ablation study on HC3 dataset. Lower is better.

| $\lambda$ | open-qa | wiki-csai | medicine | finance | Avg. |
|---|---|---|---|---|---|
| 0.001 | 0.73 | 0.72 | 0.97 | **0.79** | **0.80** |
| 0.005 | 0.72 | 0.74 | **0.96** | 0.80 | **0.80** |
| 0.01 | 0.74 | 0.74 | 0.97 | 0.80 | 0.81 |
| 0.05 | 0.73 | **0.71** | 0.97 | **0.79** | **0.80** |
| 0.1 | 0.74 | 0.74 | 0.97 | 0.81 | 0.81 |
| 0.5 | **0.71** | 0.76 | 0.97 | 0.80 | 0.81 |

We conducted ablation experiments on the explainer's hyperparameter $\lambda$ using the HC3 dataset and evaluated the interpretation results following the MoRF protocol. The results are presented in Figure 9 and Table 21. As shown, the explanation performance is largely robust to different choices of $\lambda$.

**Motifs Statistical Analysis**. We provide more statistical analysis on M4 and RAID datasets. Table 22, 24, and 23 reveal distinct motif fingerprints—frequency variations between HGT and MGT across tokens(nodes) and token-token co-occurrences(edges). Selecting the top 0.05% of edges as global explainable motifs highlights a notable difference: HGT shows a higher ratio of token and token-token co-occurrences compared to MGT. This suggests that for MGT detection, word-to-word connections are more influential than for HGT detection, given the same number of tokens. One possible explanation is that language models excel at utilizing diverse word collocations, while humans tend to rely on more conventional patterns.

Table 22: Statistics of text covered by explanation motifs on HC3 dataset. The sparsity of the explanation motifs is 0.05%.

| Statistic | open-qa | | wiki-csai | | medicine | | finance | |
|---|---|---|---|---|---|---|---|---|
| | HGT | MGT | HGT | MGT | HGT | MGT | HGT | MGT |
| Nodes | 610 | 2407 | 1685 | 777 | 923 | 990 | 1251 | 618 |
| Edges | 277 | 3496 | 2180 | 1993 | 797 | 2086 | 2004 | 1816 |
| Nodes/Edges | 2.20 | 0.69 | 0.77 | 0.39 | 1.16 | 0.47 | 0.62 | 0.34 |

Table 23: Statistics of text covered by explanation motifs on RAID dataset. The sparsity of the explanation motifs is 0.05%.

| Statistic | recipes | | book | | poetry | | review | |
|---|---|---|---|---|---|---|---|---|
| | HGTs | MGTs | HGTs | MGTs | HGTs | MGTs | HGTs | MGTs |
| Nodes | 1100 | 458 | 4093 | 1116 | 2892 | 760 | 2674 | 1560 |
| Edges | 3519 | 2567 | 8583 | 4163 | 7731 | 3452 | 5100 | 5791 |
| Nodes/Edges | 0.31 | 0.18 | 0.48 | 0.27 | 0.37 | 0.22 | 0.52 | 0.27 |

Table 24: Statistics of text covered by explanation motifs on M4 dataset. The sparsity of the explanation motifs is 0.05%.

| Statistic | wikihow | | reddit | | peerread | | arxiv | |
|---|---|---|---|---|---|---|---|---|
| | HGT | MGT | HGT | MGT | HGT | MGT | HGT | MGT |
| Nodes | 4207 | 1894 | 3929 | 1282 | 2138 | 609 | 1449 | 725 |
| Edges | 19511 | 8819 | 8448 | 2770 | 10937 | 6044 | 2954 | 2112 |
| Nodes/Edges | 0.22 | 0.21 | 0.47 | 0.46 | 0.20 | 0.10 | 0.49 | 0.34 |

**Visualizations**. To visualize the extracted motifs, we utilized the PubMed dataset, which includes MGT samples generated by three LLMs: GPT-4, Claude-3, and Davinci. We present the identified motifs at two levels of granularity: individual words and multi-word phrases or even entire sentences. We specifically extracted word-level motifs from one-hop neighbor subgraphs to visualize word-level motifs. As shown in Table 25, we selected the top 20% of tokens based on their motif scores for visualization. Similarly, for visualizing higher-level motifs (phrases/sentences) in Table 26, we extracted them from two-hop subgraphs, with the top-k ratio set to 2% for display.

# D LLM USAGE

In this paper, we leverage LLMs, including ChatGPT and Gemini 2.5 Pro, to refine sentence-level writing.

Table 25: Samples of words explanation motifs.

| Graph Motifs | Words Mapping |
|---|---|
| 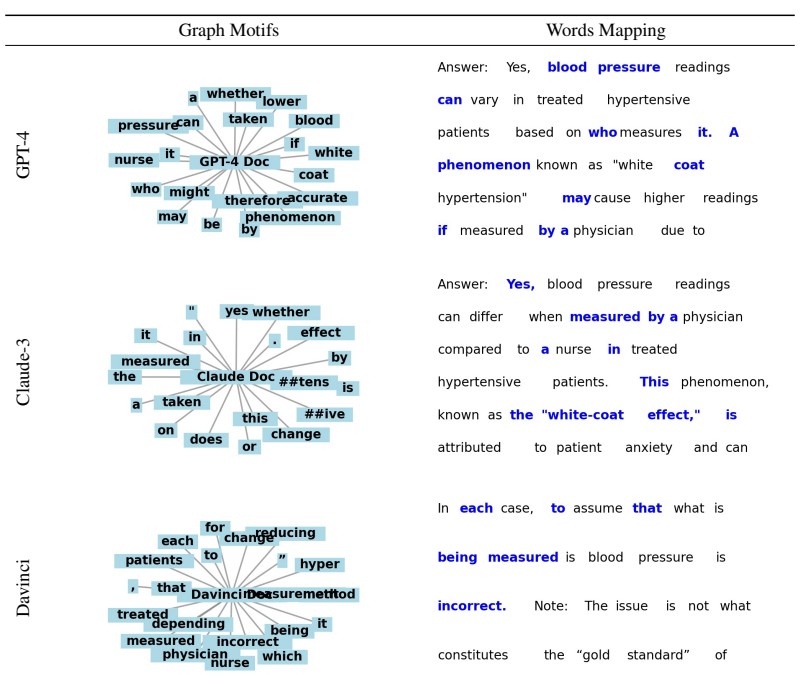 | **GPT-4** Answer: Yes, **blood pressure** readings **can** vary in treated hypertensive patients based on **who** measures **it. A phenomenon** known as "white **coat** hypertension" **may** cause higher readings **if** measured **by a** physician due to |
| | **Claude-3** Answer: **Yes,** blood pressure readings can differ when **measured by a** physician compared to **a** nurse **in** treated hypertensive patients. **This** phenomenon, known as **the** "white-coat **effect," is** attributed to patient anxiety and can |
| | **Davinci** In **each** case, **to** assume **that** what is **being measured** is blood pressure is **incorrect.** Note: The issue is not what constitutes the "gold standard" of |

Table 26: Samples of phase explanation motifs.

| Graph Motifs | Words Mapping |
|---|---|
| 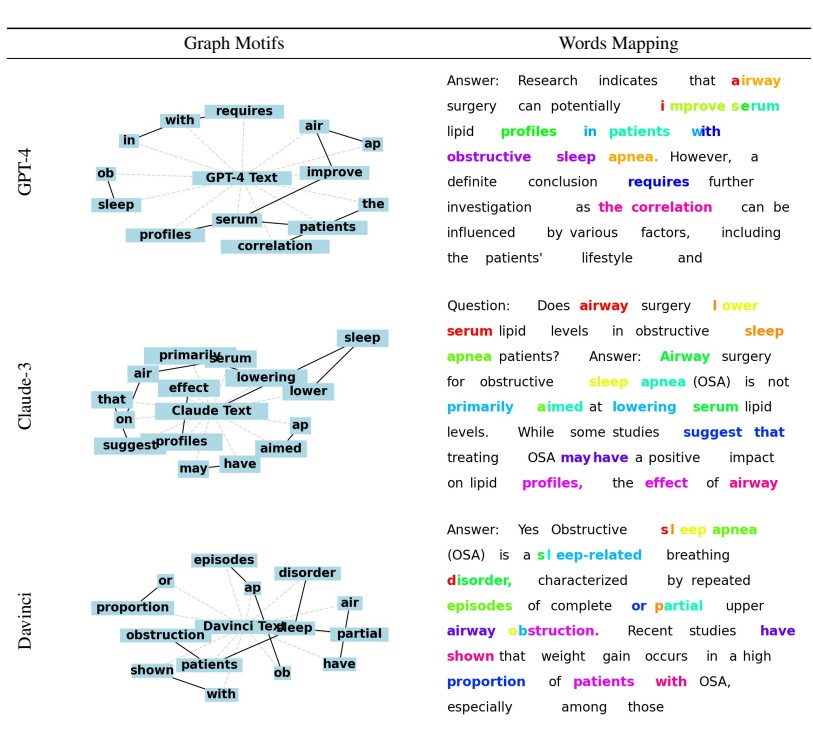 | **GPT-4** Answer: Research indicates that **a**irway surgery can potentially **i**mprove s**e**rum lipid **profiles in patients with obstructive sleep apnea.** However, a definite conclusion **requires** further investigation as **the correlation** can be influenced by various factors, including the patients' lifestyle and |
| | **Claude-3** Question: Does **airway** surgery **l**ower **serum** lipid levels in obstructive **sleep apnea** patients? Answer: **Airway** surgery for obstructive **sleep apnea** (OSA) is not **primarily a**imed at **lowering serum** lipid levels. While some studies **suggest that** treating OSA **may have** a positive impact on lipid **profiles,** the **effect** of **airway** |
| | **Davinci** Answer: Yes Obstructive **sl**eep **apnea** (OSA) is a **sl**eep-related breathing **d**isorder, characterized by repeated **episodes** of complete **or partial** upper **airway o**bstruction. Recent studies **have shown** that weight gain occurs in a high **proportion** of **patients with** OSA, especially among those |