# OpenReview forum: "LM$^2$otifs: An Explainable Framework for Machine-Generated Texts Detection"
_ICLR.cc/2026/Conference — Submitted to ICLR 2026_

### Official Review · Reviewer_U3Bx · 2025-10-29

**Soundness:** 3
**Presentation:** 2
**Contribution:** 1
**Rating:** 2
**Confidence:** 2

**Summary:**

This paper proposes an explainable framework that specifically focuses on machine-generated text detection, which brings together two existing approaches: TextGCN and GNNExplainer. Specifically, the framework first builds a token-document graph, then trains a GNN detector for machine-generated texts; GNNExplainer is positioned as a post-hoc process that additionally explains the framework’s decisions at inference time by highlighting motifs relevant to those decisions. The authors conducted extensive experiments across various test settings and compared the performance of the proposed method with a large set of SOTA MGT detectors.

**Strengths:**

- This paper is properly organized, with a clear task statement and an illustration of the proposed pipeline, which facilitates easier understanding.
- The scale of the experiments is large. The experiments bring together different detectors and test them under various settings, reflecting the SOTA in MGT detection.

**Weaknesses:**

- The novelty raises a significant concern. This paper presents a typical A+B work, which reassembles existing approaches for a specific task.
    - The framework directly adopts TextGCN as the core for detection. Apart from the simplification of the graph by binarizing the edges, equations (2) and (3) appear largely overlapping with equations (3) and (8) in the TextGCN paper.
    - The title and the claimed contribution emphasize “explainable” as a main contribution of this work, yet the framework directly adopts an existing explanation tool (GNNExplainer) with a minor modification of the regularization term. This modification appears trivial, as it does not affect or appear in the subsequent discussion, and the use of $\lambda$ is overwritten by a different concept in Section 4.
- Some details about the GNN detector are either unexplained or confusing.
    - The detailed functionalities and implementations of AGG and COMBINE are not clear.
    - The relationship between $\boldsymbol{H}$ and $\boldsymbol{Z}$ is confusing. They are supposed to have different dimensionalities and therefore cannot be connected directly through a softmax.
    - See questions for further details.
- Explainability, as a central claim, is underrepresented. The authors leverage GNNExplainer, which appears rather isolated from the detection part. The missing connection between the two components harms the consistency of the paper as a whole.
- Some experimental settings are not clarified (see more details in questions). The claimed results that compare to other explanation baselines are missing (Figure 3 only shows the random baseline).

**Questions:**

- What is the novelty or contribution beyond the knowledge of TextGCN’s capability to perform text classification?
- What are the AGG and COMBINE functions? How do they alter/update node representations?
- What is the relationship between $\boldsymbol{H}$ and $\boldsymbol{Z}$? Let $d$ be the dimensionality of the node embedding. It is confusing how a softmax function can transform $\boldsymbol{H}\in \mathbb{R}^d$ to the prediction probability vector $\boldsymbol{Z}\in\mathbb{R}^2$ in a binary classification task.
- The dataset information presented in the appendix does not match the sources. Could the authors clarify the reason for downsampling the entries and how that is implemented?
- Could the authors explain the mismatch of the performance by DeTeCtive on M4? The original paper reports an accuracy of ~98% (can be estimated from the reported recall and F1).
- What are the training settings for the competitors? Are they fine-tuned for each test case? If the answer is positive, what does the “*” symbol indicate in the table?

---

> ### Author Response · Authors · 2025-11-20
> **Response to reviewer U3Bx(Part 1/2)**
>
> Dear Reviewer U3Bx,
>
> Thank you very much for your valuable feedback. We appreciate your time and effort. Our responses are provided below.
>
> > W1. The novelty raises a significant concern.
>
>
> We appreciate the reviewer’s comment regarding innovation. In this paper, we utilize existing techniques, but these are not the core contributions of our work. One of our key contributions is highlighting the lack of support in current machine-generated text detection methods, which is important for applications such as authorship verification. **For example, in ICLR 26, the use of AI to generate comments garnered significant attention[(link)](https://x.com/gneubig/status/1989681438577336401?s=20). Although existing tools exist for detection, these tools do not provide explicit evidence to support the detection results.** In the introduction, we note that although existing human–machine text detection methods achieve good performance, explainable methods in this area remain under-explored.
>
> Our work introduces a framework from a probabilistic graphical model (PGM) perspective and provides theoretical analysis demonstrating the advantages of graph-based methods. We implement a simple pipeline using existing techniques to verify the effectiveness of the framework. Importantly, the modules in our framework are flexible and can be replaced with alternative techniques; for example, the GNN component can be GCN, GIN, GraphSAGE, etc., and the explainer can be GNNExplainer, PGExplainer, PGMExplainer, and so on.
>
> For the hyperparameter $\lambda$, we use the default value $0.05$. To address how to select $\lambda$, we conduct an ablation study of $\lambda$ on HC3 dataset following the MORF protocol to assess explanation importance. We report the average results across 13 top-k settings [0.001，0.005，0.01，0.05，0.1，0.2，0.3，0.4，0.5，0.6，0.7，0.8，0.9] in the table below. Detailed results can be found in Figure 9 and Table 21. Under the MORF protocl, lower is better. As shown in the results, the average score has an increasing trend as the $\lambda$ increases, which is reasonable. Because $\lambda$ balances the importance weights and the prediction results of the explanation. As $\lambda$ increases, the importance weight dominates, leading to reduced effectiveness of the explanations.
>
> | $\lambda$ | open-qa | wiki-csai | medicine | finance | Avg. |
> | :---: | :---: | :---: | :---: | :---: | :---: |
> | 0.001 | 0.73 | 0.72 | 0.97 | **0.79** | **0.80** |
> | 0.005 | 0.72 | 0.74 | **0.96** | 0.80 | **0.80** |
> | 0.01 | 0.74 | 0.74 | 0.97 | 0.80 | 0.81 |
> | 0.05 | 0.73 | **0.71** | 0.97 | **0.79** | **0.80** |
> | 0.1 | 0.74 | 0.74 | 0.97 | 0.81 | 0.81 |
> | 0.5 | **0.71** | 0.76 | 0.97 | 0.80 | 0.81 |
>
> We again thank the reviewers for the helpful suggestions and have revised the corresponding parts of the paper accordingly.
>
> > W2. Some details about the GNN detector are either unexplained or confusing
>
> Thanks for your constructive comment. Following previous work [1], we use the AGG and COMBINE functions, which correspond to aggregation and concatenation operations. Here, $H$ denotes the node embedding, while $Z$ represents the prediction probabilities as mentioned in Section 3.2.
> In general, the $H$ varies across layers and is determined by predefined hyperparameters. For example, the hidden layer dimension could be [input_dim,64,2], then the final dimension of $H$ is 2, aligned with the number of classes. In other words, the final dimension is the same with the prediction class numbers. To improve clarity, we will revise the paper and add the relevant citations.
>
> > W3. Explainability, as a central claim, is underrepresented. The authors leverage GNNExplainer, which appears rather isolated from the detection part. The missing connection between the two components harms the consistency of the paper as a whole.
>
> Thank you for the insightful comments. One of the goals of this paper is to introduce a general pipeline that can both detect machine-generated text and provide supporting explanations. The Explainer functions as a post-hoc method, and its operation depends on the underlying detection architecture. We appreciate the comment and will revise the paper to clarify the relationship between these two components and emphasize their close connection.
>
> > Q1. What is the novelty or contribution beyond the knowledge of TextGCN’s capability to perform text classification?
>
> Thanks for the insightful question. As mentioned in the W1. Our contributions can be summarized in three parts. First, we identify and highlight the problem that support is missing in existing machine-generated text detection methods; second, we provide a pipeline to solve this problem. We use the existing techniques, but these are not the core contribution of this paper. Other similar techniques could replace these techniques. We appreciate the comment and will revise this section in the paper to make these points clearer.

---

> ### Author Response · Authors · 2025-11-20
> **Response to reviewer U3Bx(Part 2/2)**
>
> > Q2. What are the AGG and COMBINE functions? How do they alter/update node representations?
>
> Thanks for this question. The "AGG" is the aggregation function. In GNN, there are some aggregation methods, such as mean, max. In our experiments, we use the default GCN aggregation function, $a_v^{(l)} = \text{AGG}^{(l)}({h_u^{(l-1)}}:u\in\mathcal{N}(v)) = \sum_{u \in \mathcal{N}(v)}
> \frac{1}{\sqrt{d_v d_u}} \, h_u^{(l-1)}$,where $d_v$, $d_u$ are the degreee of node $v$ and $u$. The "COMBINE" is also the default combination function, $h_v^{(l)} =
> \sigma\left(
> W^{(l)}\cdot (a_v^{(l)}+\frac{h_v^{(l-1)}}{d_v})
> \right)$. We have added reference to make it easy to understand.
>
>
> > Q3. What is the relationship between $H$ and $Z$? Let $d$ be the dimensionality of the node embedding. It is confusing how a softmax function can transform $H$ to the prediction probability vector $Z$ in a binary classification task.
>
> Thanks for the question about the details. In general, a GNN consists of several layers, and each layer can have a different embedding dimension, which is set via hyperparameters. In our experiments, the last layer dimension is the same as the number of classes. So the $H$ and $Z$ have the same dimension.
>
> > Q4. The dataset information presented in the appendix does not match the sources. Could the authors clarify the reason for downsampling the entries and how that is implemented?
>
> Thank you for pointing this out. In our experiments, we do not use the full original datasets but instead a subset of certain datasets. For HC3, M4, and RAID, we randomly sample subsets across four domains, while for Yelp, Creative, and Essay, we use the entire datasets.
> There are two reasons for downsampling. First, we follow the zero-shot protocol for testing, where some datasets contain only around 300 samples, as in Fast-DetectGPT [2]. Second, during sampling, we ensure that the training set is at most ten times larger than the test set. This represents a significant advantage compared to current baselines. Unlike zero-shot or fine-tuning methods that rely on pre-trained backbones such as GPT-2 trained on massive corpora, our model is trained from scratch and requires far fewer samples. This makes our approach particularly advantageous for detecting entirely new LLMs or generative AI systems.
>
> > Q5. Could the authors explain the mismatch of the performance by DeTeCtive on M4? The original paper reports an accuracy of ~98% (can be estimated from the reported recall and F1).
>
> Thank you for pointing out this difference. The experimental setting in our paper differs from that of the DeTeCTive paper. In DeTeCTive, the evaluation is conducted at the language level, whereas our experiments are based on both the domain and language model. As mentioned in Q4, our test set is a subset of the original dataset, which also contributes to the difference in performance. We believe that the observed differences are within a reasonable range.
>
> > Q6. What are the training settings for the competitors? Are they fine-tuned for each test case? If the answer is positive, what does the “*” symbol indicate in the table?
>
> Thank you for the detailed implementation question. In our experiments, we use the pretrained model for evaluating all baselines. For example, we adopt the OUTFOX setting when using the DeTeCTive method. We use the pre-trained model for RoBERTa-QA, which is well-trained based on the HC3 dataset.
>
> Reference:
> - [1]. How Powerful are Graph Neural Networks? ICLR, 2019
> - [2]. Fast-DetectGPT: Efficient Zero-Shot Detection of Machine-Generated Text via Conditional Probability Curvature, ICLR 2024
> - [3]. DeTeCtive: Detecting AI-generated Text via Multi-Level Contrastive Learning, NIPS, 2024
> - [4]. How Close is ChatGPT to Human Experts? Comparison Corpus, Evaluation, and Detection, arXiv, 2023
>
>
> We thank you again for your constructive comments and for your efforts to improve the quality of our paper.

---

### Official Review · Reviewer_tbGZ · 2025-10-30

**Soundness:** 3
**Presentation:** 3
**Contribution:** 2
**Rating:** 4
**Confidence:** 4

**Summary:**

The paper proposes a new method for authorship attribution for distinguishing between human and machine generated text. The task is treated as  a binary classification task. Given a set of human and machine generated texts, first the text is tokenized. A graph is created including all tokens and the documents/texts (either human or machine generated) as different types of nodes. Given an unseen text, a GNN has been trained to predict the node type (human or machine) of the text by first adding any new tokens and connecting the unknown text with all its token nodes. Empirical results on standard benchmarks show that the proposed method is more effective compared to other classification methods.

**Strengths:**

- The proposed method seems to work well and it has interpretable properties.

**Weaknesses:**

- The paper propose a well known technique applied on this specific llm vs human generated classification task.

- The performance of the proposed method but also other methods looks saturated (e.g. achieving 100% accuracy in many cases). Either the task is solved or the datasets are too easy. This is not a criticism of the method proposed in this paper but for the whole area of llm/human authorship attribution.

**Questions:**

N/A

---

> ### Author Response · Authors · 2025-11-20
> **Response to reviewer tbGZ**
>
> Dear reviewer tbGZ,
>
> We sincerely appreciate your feedback and comments on our paper.  We provide the responses below:
>
> > W1. The paper propose a well known technique applied on this specific llm vs human generated classification task.
>
> Thank you for the constructive comment. In this paper, we employ several existing techniques, but these are not the core contributions of our work. One of our key contributions is highlighting the lack of support in current machine-generated text detection methods, which is crucial for applications such as authorship verification. **For example, in ICLR 26, the use of AI to generate comments garnered significant attention[(link)](https://x.com/gneubig/status/1989681438577336401?s=20). Although existing tools exist for detection, these tools do not provide explicit evidence to support the detection results.**
> In the introduction, we emphasize that although existing human–machine text detection methods have shown promising results, explainable approaches in this area remain under-explored.
>
> Our paper introduces a unified framework from a probabilistic graphical model (PGM) perspective and provides theoretical analysis demonstrating the advantages of graph-based methods. We implement a simple pipeline using existing techniques to verify the effectiveness of the proposed framework. Importantly, the modules in our framework are flexible and can be replaced by other techniques; for example, the GNN component can be GCN, GIN, GraphSAGE, etc., and the explainer can be GNNExplainer, PGExplainer, PGMExplainer, and so on.
>
> Thanks again for the valuable comments. We have revised the paper to clarify these points more explicitly.
>
> > W2. The performance of the proposed method but also other methods looks saturated (e.g. achieving 100% accuracy in many cases). Either the task is solved or the datasets are too easy. This is not a criticism of the method proposed in this paper but for the whole area of llm/human authorship attribution.
>
> Thank you for the thoughtful comment. In previous work [1–3], the detection accuracy is already high, and even with the most recent LLMs, the accuracy remains strong. This is not because the datasets are too easy; rather, there are detectable patterns in Human-Generated Texts (HGT) and Machine-Generated Texts (MGT). However, existing methods provide limited discussion on how to offer support or explanations for their detection results.
>
> Reference:
> - [1]. Glimpse: Enabling White-Box Methods to Use Proprietary Models for Zero-Shot LLM-Generated Text Detection, ICLR 2025.
> - [2]. Zero-Shot Detection of LLM-Generated Text using Token Cohesiveness, EMNLP 2024.
> - [3]. Fast-DetectGPT: Efficient Zero-Shot Detection of Machine-Generated Text via Conditional Probability Curvature, ICLR 2024.
>
> We thank you again for your constructive comments and for your efforts to improve the quality of our paper.

---

### Official Review · Reviewer_3rdZ · 2025-10-31

**Soundness:** 3
**Presentation:** 3
**Contribution:** 3
**Rating:** 6
**Confidence:** 3

**Summary:**

The paper proposes an explainable machine-generated text detection framework based on Probabilistic Graphical Models (PGM) and eXplainable Graph Neural Networks (XGNN). It constructs word co-occurrence graphs and employs GNNs for detection, while extracting “explainable motifs” that reveal linguistic structures distinguishing human-written and machine-generated texts.

**Strengths:**

S1: The introduction of a PGM perspective into machine-generated text detection (MGT detection) provides a theoretical justification (Theorem 4.1) and represents a meaningful level of innovation.

S2: The method is comprehensively validated across six mainstream datasets (HC3, M4, RAID, Yelp, Essay, Creative) and multiple LLMs (GPT-4, Claude3, Gemini, etc.), showing wide coverage and robustness.

S3: The model achieves strong performance and high inference efficiency (testing time only 0.005–0.009s/sample). It outperforms major baselines such as DetectGPT, Binoculars, Fast-DetectGPT, and DeTeCtive on both ACC and AUC metrics.

**Weaknesses:**

W1: Although the paper claims theoretical novelty from the PGM perspective, the actual modeling is limited to co-occurrence graphs plus GCNs, making it conceptually similar to TextGCN (Yao et al., 2019) and GNNExplainer (Ying et al., 2019). The theoretical analysis lacks new mechanisms or proof-level innovations.

W2: Several methodological details are unclear. For example, in the graph construction stage, the choice of PMI threshold, sliding window size, and hyperparameter λ is not specified. In motif extraction (Eq. 4), the selection of λ and the reproducibility of the extracted subgraphs are insufficiently discussed.

W3: The model’s cross-domain performance drops significantly (e.g., only 0.59 in the Reddit domain), indicating dependence on dataset-specific statistics and limited generalization capability.

W4: Typos and formatting issues: In Section 4, several mathematical symbols are densely presented without clear definitions. Minor typos exist (e.g., in Eq. (3), “Ydℓ is the ground-truth label,.” has an extra comma). Some equations are broken across lines, and figure–table references are inconsistent (e.g., Figure 3 and Table 2). Table captions sometimes lack consistent expansion of abbreviations (e.g., “DaV.”, “Dol.” should be fully spelled out on first mention).

**Questions:**

In Eq. (4), λ controls the balance between fidelity and complexity in subgraph extraction. How stable are the resulting motifs across multiple runs or random seeds? Do the motifs remain semantically consistent, or do they vary significantly due to stochastic optimization?

---

> ### Author Response · Authors · 2025-11-20
> **Response to reviewer 3rdZ (Part 1/3)**
>
> Dear Reviewer 3rdZ,
>
> Thank you very much for taking the time to review our manuscript and providing insightful and valuable comments. Your feedback has been incredibly helpful in improving the quality of this work.
>
> > W1. Although the paper claims theoretical novelty from the PGM perspective, the actual modeling is limited to co-occurrence graphs plus GCNs, making it conceptually similar to TextGCN (Yao et al., 2019) and GNNExplainer (Ying et al., 2019). The theoretical analysis lacks new mechanisms or proof-level innovations.
>
> We appreciate your comment regarding the concerns about innovation. In this paper, we utilize existing techniques; however, these techniques are not the core contribution. One of our contributions is to highlight the lack of support in existing machine-generated text detection methods, which is important in applications such as authorship verification. **For example, in ICLR 26, the use of AI to generate comments garnered significant attention[(link)](https://x.com/gneubig/status/1989681438577336401?s=20). Although existing tools exist for detection, these tools do not provide explicit evidence to support the detection results.**
> In the introduction, we point out that although existing human–machine text detection methods achieve good performance, explainable methods in this area remain under-explored.
>
> In this paper, we propose a framework from a probabilistic graphical model perspective and provide theoretical analysis demonstrating the advantages of graph-based methods. We implement a simple pipeline using existing techniques to verify its effectiveness. Importantly, in our framework, the modules are flexible and can be replaced with alternative techniques; for example, the GNN models can be GCN, GIN, GraphSAGE, etc., and the explainer can be GNNExplainer, PGExplainer, PGMExplainer, and so on.
>
> Thanks again for the insightful comments. We have revised some parts to make these points clearer.

---

> ### Author Response · Authors · 2025-11-20
> **Response to reviewer 3rdZ (Part 2/3)**
>
> > W2. Several methodological details are unclear. For example, in the graph construction stage, the choice of PMI threshold, sliding window size, and hyperparameter λ is not specified. In motif extraction (Eq. 4), the selection of λ and the reproducibility of the extracted subgraphs are insufficiently discussed.
>
> Thanks for the valuable comments. For the PMI threshold, we do not use a threshold for PMI, effectively, the threshold is 0. In Eq. (2), an edge is added when PMI > 0 (i.e., when tokens i and j co-occur within a window). In Appendix C.1 Table 4, we have ablation study on if we should use the PMI as edge weight. The results show that the setting without PMI weights achieves the best performance.
>
> For the sliding window size, we follow the default setting 20 used in the TextGCN paper. To adress the concern about this hyperparameter, we add an additional set of experiments on HC3 dataset. As the results show in the table below, our method is robust with respect to different window sizes.
>
> | Metric | Window Size | open-qa | wiki-csai | medicine | finance | Avg. |
> | :--- | :---: | :---: | :---: | :---: | :---: | :---: |
> | **ACC** | 10 | 0.95 | 0.93 | 0.99 | 0.93 | 0.95 |
> | | 15 | 0.97 | 0.95 | 0.99 | 0.94 | 0.96 |
> | | 20 | 0.97 | **0.96** | 0.98 | **0.98** | **0.97** |
> | | 25 | 0.96 | 0.94 | **1.00** | 0.94 | 0.96 |
> | | 30 | **0.98** | 0.93 | 0.99 | 0.93 | 0.96 |
> | **AUC** | 10 | 0.99 | 0.99 | **1.00** | 0.98 | 0.99 |
> | | 15 | **1.00** | 0.99 | **1.00** | 0.98 | 0.99 |
> | | 20 | **1.00** | 0.99 | **1.00** | **1.00** | **1.00** |
> | | 25 | **1.00** | **1.00** | **1.00** | 0.98 | **1.00** |
> | | 30 | **1.00** | **1.00** | **1.00** | 0.98 | **1.00** |
>
> For the hyperparameter $\lambda$, we use the default setting $0.05$. To adress how to select the $\lambda$, we conduct an ablation study of $\lambda$ on HC3 dataset following the MORF protocol to assess explanation importance. We report the average results across 13 top-k settings [0.001，0.005，0.01，0.05，0.1，0.2，0.3，0.4，0.5，0.6，0.7，0.8，0.9] in table below. Detailed results can be found in Figure 9 and Table 21. Under MORF protocl, lower is better. As shown in the results, the average score have a increasing trend as the $\lambda$ increase, which is reasonable. Because $\lambda$ balances the importance weights and the prediction results of the explanation. As $\lambda$ increases, the importance weight dominates, leading to reduced effectiveness of the explanations.
>
> | $\lambda$ | open-qa | wiki-csai | medicine | finance | Avg. |
> | :---: | :---: | :---: | :---: | :---: | :---: |
> | 0.001 | 0.73 | 0.72 | 0.97 | **0.79** | **0.80** |
> | 0.005 | 0.72 | 0.74 | **0.96** | 0.80 | **0.80** |
> | 0.01 | 0.74 | 0.74 | 0.97 | 0.80 | 0.81 |
> | 0.05 | 0.73 | **0.71** | 0.97 | **0.79** | **0.80** |
> | 0.1 | 0.74 | 0.74 | 0.97 | 0.81 | 0.81 |
> | 0.5 | **0.71** | 0.76 | 0.97 | 0.80 | 0.81 |
>
> For reproducibility, we conduct experiments using two random seeds and calculate the overlap ratio. We report the average and error bars on the HC3 dataset across different top-k settings[0.1,0.2,0.3,0.4,0.5]. As shown in the table below, our method demonstrates high reproducibility.
>
> | Sparsity | open-qa | wiki-csai | medicine | finance |
> | :---: | :---: | :---: |  :---: | :---: |
> |0.1| 0.9730 $\pm$ 0.0052 | 0.7888 $\pm$ 0.0697 | 0.8510 $\pm$ 0.0733| 0.7876 $\pm$ 0.0822 |
> |0.2| 0.9858 $\pm$ 0.0033 | 0.9056 $\pm$ 0.0276 | 0.9241 $\pm$ 0.0642| 0.9058 $\pm$ 0.0325 |
> |0.3| 0.9900 $\pm$ 0.0024 | 0.9274 $\pm$ 0.0226 | 0.9495 $\pm$ 0.0650| 0.9337 $\pm$ 0.0231
> |0.4| 0.9946 $\pm$ 0.0013 | 0.9501 $\pm$ 0.0147 | 0.9678 $\pm$ 0.0655| 0.9562 $\pm$ 0.0154 |
> |0.5| 0.9967 $\pm$ 0.0006 | 0.9703 $\pm$ 0.0084 | 0.9780 $\pm$ 0.0664| 0.9724 $\pm$ 0.0091 |
>
> We again thank the reviewers for the helpful suggestions and have revised the corresponding parts of the paper accordingly.

---

> ### Author Response · Authors · 2025-11-20
> **Response to reviewer 3rdZ (Part 3/3)**
>
> > W3. The model’s cross-domain performance drops significantly (e.g., only 0.59 in the Reddit domain), indicating dependence on dataset-specific statistics and limited generalization capability.
>
> We thank the reviewer for the insightful comments.  Our pipeline is trained on a much smaller number of samples compared to other LLM-based methods such as DetectGPT, which rely on large, well-trained language models.  The cross-domain performance naturally depends on the distribution differences between corpora from different domains.
>
> The result on reddit is obtained using a model pretrained on wiki-how, which indicates that the distributions of reddit and wiki-how differ significantly. Because in some cases, such as the peerread and arxiv, the cross-domain results are relative comparable.  We believe the cross-domain performance can be improved by adding training data with target domains. However, as the LLMs spring up, training a robust detector for all domains would require substantial time and money. To build a detector for a new LLM in some domain, our method has advantage over existing approaches.
>
> Thanks again foir the insightful comments. We consider improving cross-domain generalization as an important direction for future work.
>
>
> > W4. Typos and formatting issues: In Section 4, several mathematical symbols are densely presented without clear definitions. Minor typos exist (e.g., in Eq. (3), “Ydℓ is the ground-truth label,.” has an extra comma). Some equations are broken across lines, and figure–table references are inconsistent (e.g., Figure 3 and Table 2). Table captions sometimes lack consistent expansion of abbreviations (e.g., “DaV.”, “Dol.” should be fully spelled out on first mention).
>
> Thanks for pointing out these issues. We appreciate your time and efforts. We will double-check our paper and fix them in the next version.
>
>
> > Q1. In Eq. (4), λ controls the balance between fidelity and complexity in subgraph extraction. How stable are the resulting motifs across multiple runs or random seeds? Do the motifs remain semantically consistent, or do they vary significantly due to stochastic optimization?
>
> Thanks for the insightful questions. As the results show in W2, our method demonstrates high reproducibility with different random seed. In additional, the subgraph extraction reproducibility is also determinated by the explainer. If we utilize a gradient-based method, the subgraph of two random seed would be perfect aligned.
>
> We thank you again for your constructive comments and for your efforts to improve the quality of our paper.

---

### Official Review · Reviewer_MRb9 · 2025-11-01

**Soundness:** 3
**Presentation:** 2
**Contribution:** 2
**Rating:** 4
**Confidence:** 3

**Summary:**

The paper introduces LM^2OTIFS, an explainable MGT detection framework, inspired by probabilistic graphical models (PGM).
The framework uses the word cooccurrence to capture the lexical dependence; then it applies XGNNs to learn to predict and generate motifs for explanation.
The framework outperforms baselines in terms of interpretability, with competitive performance.

**Strengths:**

1. The idea of linking the explanability in the graph neuron network to explain machine-generated text detection is impressive.
2.  The experiment covers adequate baseline methods and datasets.

**Weaknesses:**

1. The explanation results have space to be expanded and deepened. 1) Unclear how to explain the difference between human and machine text. For the example in Figure 4, I believe "sleep" and "patients" are also very common words in human-written medical-related documents. It is still unclear why these words are a strong clue. 2) There are also highlighted non-common words, e.g., "in", "with". They may indicate more generalizable patterns, but the paper fails to discuss.  3) My understanding of the current explanation is limited to the cooccurrence of words, which might be highly conditioned on the training corpus. I sense why "obstructive sleep" is a matify for GPT-4 might just be because it has appeared as a training example (which I can be wrong). But what if we remove those "obstructive sleep" examples as a small perturbation of the dataset collection? Can the detector still be robust on the test example?

2. I am concerned about the computation cost. It seems each time we predict a new document, we need to compute its tokens' occurrence with all training dataset (I can be wrong), which could be a large cost when we scale up the training set and test set. But for other baselines, the estimated distribution is mostly parametrically stored, which only requires a pass on the to-be-predicted document.

3. There are spaces to improve writing clarity. Sections 3.1 and 4 seem to have large overlap content on the graph construction/definition parts.

4. (A minor point) The explanation of detection is limited to the lexical level. As token nodes are initialized with one-hot features and sequence nodes with all-zero features, I would suggest that no semantic information is introduced. As some papers point out that semantics is important for some MGT detection (like some pos types of tokens may contribute more), it would be cool to see if we include semantic features, for example, initializing the graph node with embeddings from some LLM, the motifs would be the same or not.

**Questions:**

1. Section 2 MGT Detection: Should T_h and T_m be an input to the function f for the tasks? It seems to include the few-shot detectors. I assume you are mainly considering your methods, but what about all your other baselines? Do you mean they can all also be expressed as this function?

---

> ### Author Response · Authors · 2025-11-20
> **Response to reviewer MRb9 (Part 1/2)**
>
> Dear Reviewer MRb9,
>
> Thank you very much for your valuable feedback. We appreciate your time and efforts. Our responses are provided below.
>
> > W1.1 & W1.2 Unclear how to explain the difference between human and machine text.  There are also highlighted non-common words, e.g., "in", "with". They may indicate more generalizable patterns, but the paper fails to discuss.
>
>
> Thank you for the thoughtful comment. The prediction decision depends on the complex combinations of words. In Figure 4, we visualize the most influential parts. There are several reasons why seemingly common words can still be important.
>
> First, their importance can arise from their distribution or frequency within the answers. For example, watermark-like signals can influence the model’s probability over certain words even when the sentences appear similar.
> If a word that appears frequently in human-written text, but rarely in LLM-generated text due to safety or forbidden-word filters, is present at high frequency in a sentence, it increases the likelihood that the sentence is human-written.
> Second, these common words may form stronger patterns when combined with other words. In Figure 4, we highlight important co-occurrence structures that strongly affect the prediction. These co-occurrences can capture stylistic or structural signals that individual words alone cannot represent.
>
> Thank you again for the helpful comments. We revised this part of the paper and provide additional discussion and explanation to make it easier to understand.
>
>
>
>
>
> > W1.3 My understanding of the current explanation is limited to the cooccurrence of words, which might be highly conditioned on the training corpus. I sense why "obstructive sleep" is a matify for GPT-4 might just be because it has appeared as a training example (which I can be wrong). But what if we remove those "obstructive sleep" examples as a small perturbation of the dataset collection? Can the detector still be robust on the test example?
>
> Thank you for the insightful comments and questions. It is true that our method relies on the training corpus. During training, the GNN model learns to capture distributional differences between Human-Generated Text (HGT) and Machine-Generated Text (MGT). Although the important words indeed appear in the training set, their importance arises from the differences in how they are distributed across HGT and MGT. As in the watermark example, each word has some probability of appearing, but what matters for detection is not the presence of a single word, it is the overall distribution pattern and the co-occurrence structure.
>
> If these patterns are removed, the detector would not be able to make correct predictions. The decision-making process often depends on complex combinations of words and their co-occurrences, so removing just one or two words typically has limited influence on the final prediction.
>
> This aspect relates to the evaluation of explanations. For example, in Figure 3, when important attributions are removed, the prediction accuracy drops significantly, whereas removing unimportant attributions has minimal effect. This demonstrates that the model relies on these distributional and structural patterns rather than isolated tokens.
>
>
> > W2. I am concerned about the computation cost. It seems each time we predict a new document, we need to compute its tokens' occurrence with all training dataset (I can be wrong), which could be a large cost when we scale up the training set and test set. But for other baselines, the estimated distribution is mostly parametrically stored, which only requires a pass on the to-be-predicted document.
>
> Thank you for raising the concern about computation cost. We discuss this issue in Appendix C.1, where we note that the additional cost comes from graph construction, whose time complexity is $O(LW^2)$, where $L$ is the length of sentence and $W$ is the size of sliding window.  In practice, for a new document, the only required operation is to extract the words and connect the document node to the corresponding word nodes.
>
> To make this clearer, we view graph construction during training as consisting of two stages. First, we extract all words as word nodes and construct their co-occurrence relationships as word–word edges. In the second stage, we add document nodes and connect them to the relevant word nodes. During test, we only need to repeat the second stage, adding new document nodes and their doc–word edges. This second stage is lightweight and efficient.
>
> Thank you again for pointing out this concern. We have revised the paper to clarify this part.

---

> ### Author Response · Authors · 2025-11-20
> **Response to reviewer MRb9 (Part 2/2)**
>
> > W3. There are spaces to improve writing clarity. Sections 3.1 and 4 seem to have large overlap content on the graph construction/definition parts.
>
> Thank you for the valuable suggestion. We have double checked and reorganize the sections to make it clear.
>
>
> > W4. The explanation of detection is limited to the lexical level. As token nodes are initialized with one-hot features and sequence nodes with all-zero features, I would suggest that no semantic information is introduced. As some papers point out that semantics is important for some MGT detection (like some pos types of tokens may contribute more), it would be cool to see if we include semantic features, for example, initializing the graph node with embeddings from some LLM, the motifs would be the same or not.
>
>
> Thanks for the insightful comments. In this paper, we follow the standard graph-preprocessing procedure and initialize node features using one-hot representations. To address the concern regarding the lack of semantic information in such features, we conducted an additional experiment using Bert token embeddings as node features. As shown in the table below, incorporating Bert embeddings does not lead to performance improvements.
>
> We attribut this to the high similarity of tokens in semantic embedding space. As we mentioned in W1.1 & W1.2, LLM could have a forbidden/permitted list for pairs of words by using the watermark techique, which have similar semantic feature embedding. When using semantic embeddings alone, these similarities make it difficult for the detector to learn the discriminative patterns between HGT and MGT.
>
> | Metric | Method | open-qa | wiki-csai | medicine | finance | Avg. |
> | :--- | :--- | :---: | :---: | :---: | :---: | :---: |
> | **ACC** | Ours | 0.97 | 0.96 | 0.98 | **0.98** | **0.97** |
> | | Ours-Bert | **1.00** | 0.79 | 0.98 | 0.89 | 0.91 |
> | **AUC** | Ours | **1.00** | 0.99 | **1.00** | **1.00** | **1.00** |
> | | Ours-Bert | **1.00** | 0.85 | 0.99 | 0.97 | 0.95 |
>
>
>
> > Q1.  Should T_h and T_m be an input to the function f for the tasks? It seems to include the few-shot detectors. I assume you are mainly considering your methods, but what about all your other baselines? Do you mean they can all also be expressed as this function?
>
> Thank you for insights of this part. In section 2, $\mathcal{T}_h$ and $\mathcal{T}_m$ denote the paired training set. To provide a universal formulation, we use $f:(\mathcal{T}_h,\mathcal{T}_m,S_o) \mapsto \widehat{Y}$ to represent the detection mechanism.  This formulation does not assume how the training sets are used.
> For the training-based methods, they can use them to train the function. For these methods based on the well-trained LLM models, the training set may not be used at all.  The formulation is designed to unify both types of approaches.
>
> In our experiments, we include a variety of baselines covering both training-based and zero-shot detection methods. Detailed descriptions of these methods are provided in the paper.
>
>
> We thank you again for your constructive comments and for your efforts to improve the quality of our paper.

---

> > ### Comment · Reviewer_MRb9 · 2025-11-28
> > **Thank you**
> >
> > I thank the authors for their response. I will adjust my assessment accordingly.
> >
> > To Area Chairs:
> >
> > I hope to adjust my assessment accordingly, but I found I don't have edit access for now. Can you help check it? Thank you!

---

### Author Response · Authors · 2025-11-26
**Global Response to Reviewers**

Dear Reviewers,

We thank all reviewers for the constructive and valuable feedback. In response, we have made substantial revisions to improve the clarity, structure, and empirical validation of our work. Below, we summarize the major updates incorporated into the revised manuscript:

- We clarify that one of our key contributions is highlighting the problem of missing evidence support in MGT detection.

- We have reorganized the Theoretical Analysis and Methodology sections to improve logical flow. In the Preliminaries, we now clearly explain that $\mathcal{T}_h$ and $\mathcal{T}_m$ refer to support samples rather than inputs to the detection function.

- We have clarified our graph construction pipeline as a two-stage process: (1) building token–token graphs, and (2) adding document/sentence nodes and connecting them with token nodes. We have also added appropriate citations for the GNN modules used.

- For the qualitative analysis, we provide a more detailed explanation of why certain words and structures are important. To better illustrate this, we use watermarking techniques as an example to show why even subtle similarities in text or structure can be crucial for the classifier.

- We have added an ablation study on window size and the hyperparameter $\lambda$ in Appendix C.

We are grateful for the reviewers’ feedback, which has significantly improved the quality of our work. We hope that the clarifications and additional analyses provided above will assist in your final assessment of our submission.

---

### Author Response · Authors · 2025-11-27
**Looking forward to your reply**

Dear Reviewers,

We have carefully considered all reviewer comments and made substantial revisions to both the response and the updated manuscript. To facilitate the review process, we have also summarized the key changes and provided additional clarifications in the revision.

As the review deadline is approaching, we would greatly appreciate any feedback you may have on the revised submission at your earliest convenience. Your insights are invaluable to us, and we sincerely welcome any further comments or suggestions.

Thank you very much for your time and effort in reviewing our work. Please feel free to let us know if any additional clarification or information would be helpful.

Best,

The Authors

---

### Author Response · Authors · 2025-12-02
**Summary of Rebuttal and Contributions**

Dear Area Chair,

We sincerely appreciate your dedication to evaluating our submission and rebuttal. The additional reviewing responsibilities undertaken by Area Chairs are invaluable, and we acknowledge the complexities of ensuring equitable assessments under such demanding conditions. We respectfully submit for your consideration: (1) a synopsis of our contributions, (2) an overview of how our rebuttal addressed reviewers' main concerns.

**Main contribution of this paper:**

- We highlight the issues that the existing MGT detection methods lack evidence support. We provide a new explainable framework from the Probabilistic Graphical Models aspects, which provides the detection and support explanation.
- We provide a theoretical analysis of the rationale and advantages of employing a Graph Neural Network(GNN) for this task, drawing insights from the perspective of PGM.
- We conduct comprehensive experiments across multiple datasets to validate both the detection performance and the framework’s ability to extract meaningful explanatory motifs.


**Address the Reviewers' Main Concerns:**

1.  **_Novelty & Contribution(Reviewers: 3rdZ, tbGZ, U3Bx)_**

     The key contribution of this paper is not new techniques or modules. Our contributions are in three aspects:
      - Our contribution highlights the evidence-support gap in existing methods. For example, in ICLR 26, the use of AI to generate comments garnered significant attention[(link)](https://x.com/gneubig/status/1989681438577336401?s=20), although existing tools exist for detection. However, these tools do not provide explicit evidence to support the detection results.
       - Our paper introduces a unified framework from a probabilistic graphical model (PGM) perspective and provides theoretical analysis demonstrating the advantages of graph-based methods.  Importantly, the modules in our framework are flexible and can be replaced with alternative techniques; for example, the GNN component can be GCN, GIN, GraphSAGE, etc., and the explainer can be GNNExplainer, PGExplainer, PGMExplainer, and so on.
       - We provide a theoretical analysis and comprehensive experiments to verify the effectiveness of the detection and the ability of explainable motifs extraction.


2.  **_Explainability (MRb9, U3Bx)_**

     **Why words and co-occurrence are important.**   Prediction decisions often depend on complex word interactions. Their importance may come from distributional patterns or frequency within human-generated vs. machine-generated text. Certain combinations can amplify stylistic or “watermark-like” signals. For example, words that frequently occur in human-written text but appear less often in LLM-generated text (due to safety filters or generation biases) can produce detectable patterns that increase the likelihood of human authorship. These motivate our focus on word–word and word–context relationships.


3.  **_Methodological Details & Hyperparameters(3rdZ, MRb9, U3Bx)_**

      We have revised some parts and clarified our settings. We have added new experiments showing the method is robust to window sizes (10–30) and analyzed the impact of $\lambda$ on explanation quality. For the computational cost, we now clearly describe the two-stage graph construction. Stage 1 (Word-Word edges) happens during training. Stage 2 (Doc-Word edges) is the only step required at inference, making it efficient.


We are grateful for the reviewers’ feedback, which has significantly improved the quality of our work. We are encouraged that Reviewer MRb9 responded positively to these clarifications and indicated an intention to adjust the assessment.

Thank you again for your time and careful consideration of our work. We hope this summary, our revised manuscript, and rebuttal will be helpful for your evaluation.

Best,
The Authors

---

### Meta-Review · Area_Chair_cG3f · 2026-01-03

**Summary:**

The following five types of concerns can be summarized below:

(1) Novelty Concerns. Reviewers questioned the novelty of the proposed framework, noting that it largely assembles existing techniques, TextGCN for graph construction, and GNNExplainer for post-hoc explanation, without introducing fundamentally new mechanisms or proof-level innovations. Some reviewers argued it is a typical "A+B" work, and the core contributions are not sufficiently distinct from prior research, as the explanation component relies on existing tools with minor modifications.

(2) Limited Explainability. While the framework emphasizes explainability as a core contribution, reviewers found the explanation results underdeveloped. Criticisms include unclear reasoning for why common words or specific word co-occurrences serve as discriminative clues, over-reliance on lexical-level co-occurrence without incorporating semantic information, and a lack of connection between the detection and explanation.

(3) Efficiency Concerns. Concerns about computational efficiency were raised, with questions about the cost of graph construction when scaling training/test sets.

(4) Generalization Limitations. Some reviewers pointed out that existing methods show near-saturated performance on benchmarks, suggesting that datasets may be too easy, and the framework’s robustness to training data perturbations is unproven. I agree with these reviewers and suggest that authors use more challenging LLMs, such as Gemini 3.0 and Qwen 3.5. Reviewers also noted significant cross-domain performance drops (e.g., accuracy of 0.59 in the Reddit domain when trained on wiki-how), indicating dependence on dataset-specific statistics and limited generalization.

(5) Writing Issues. Overlapping content between sections, and formatting/typo issues, e.g., undefined mathematical symbols, inconsistent figure-table references.

**Reviewer Concerns:**

The rebuttal may address the reviewer's concerns:

- For novelty concern, authors clarified that the core contribution lies in highlighting the "evidence-support gap" in existing MGT detection methods and proposing a flexible, unified framework from a PGM perspective.

- For methodological details, authors supplemented ablation studies on hyperparameters, clarified graph construction as a two-stage process, defined AGG/COMBINE functions, and committed to fixing formatting/typo issues.

- For efficiency, authors clarified that inference only requires the lightweight second stage of graph construction (doc-word edges), with time complexity O(LW²), addressing scaling concerns.

The reviewer's concerns may still be outstanding:

- Novelty remains questionable, as the framework relies on existing techniques without introducing new mechanisms; the "evidence-support gap" highlight is seen as an incremental contribution rather than a groundbreaking innovation.

- Cross-domain generalization is still unresolved, as authors acknowledged it depends on target domain training data and requires substantial resources to improve, with no immediate solutions provided.

- Explainability limitations persist: semantic information integration showed no benefit, but reviewers still believe semantics could matter for certain MGT detection scenarios; the connection between detection and explanation components is only partially clarified.

- Robustness to training data perturbations is unproven, as the authors did not conduct experiments on datasets with specific co-occurrence examples removed.

- Saturated benchmark performance raises questions about the task’s difficulty, which the authors did not address beyond attributing high accuracy to inherent HGT/MGT patterns.

**Reviewer Scores:**

- For Reviewer MRb9, may adjust the scores, as authors addressed key concerns and the reviewer indicated an intention to revise their assessment.
- For Reviewer 3rdZ, may keep the score unchanged, as authors addressed methodological details, but cross-domain generalization and theoretical innovation concerns remain outstanding.
For Reviewer tbGZ, may keep the score unchanged, as authors clarified contributions but did not address the core concern of saturated task performance.
For Reviewer U3Bx, may not change the score, as major concerns were only partially addressed, and the framework is still viewed as a derivative "A+B" work.

---

### Decision · Program_Chairs · 2026-01-26

Reject